# Formation of ribbed bedforms below shear margins and lobes of palaeo-ice streams

Jean Vérité[1], Édouard Ravier[1], Olivier Bourgeois[2], Stéphane Pochat[2], Thomas Lelandais[1], Régis Mourgues[1], Christopher D. Clark[3], Paul Bessin[1], David Peigné[1], Nigel Atkinson[4]

[1] Laboratoire de Planétologie et Géodynamique, UMR 6112, CNRS, Le Mans Université, Avenue Olivier Messiaen, 72085 Le Mans CEDEX 9, France
[2] Laboratoire de Planétologie et Géodynamique, UMR 6112, CNRS, Université de Nantes, 2 rue de la Houssinière, BP 92208, 44322 Nantes CEDEX 3, France
[3] Department of Geography, University of Sheffield, Sheffield, UK
[4] Alberta Geological Survey, 4th Floor Twin Atria Building, 4999-98 Ave. Edmonton, AB, T6B 2X3, Canada

*Correspondence to*: Jean Vérité (jean.verite@univ-lemans.fr)

## Abstract

Conceptual ice stream landsystems derived from geomorphological and sedimentological observations provide constraints on ice-meltwater-till-bedrock interactions on palaeo-ice stream beds. Within these landsystems, the spatial distribution and formation processes of ribbed bedforms remain unclear. We explore the conditions under which these bedforms may develop and their spatial organisation with (i) an experimental model that reproduces the dynamics of ice streams and subglacial landsystems and (ii) an analysis of the distribution of ribbed bedforms on selected examples of palaeo-ice stream beds of the Laurentide Ice Sheet. We find that a specific kind of ribbed bedforms can develop subglacially through soft bed deformation, where the ice flow undergoes lateral or longitudinal velocity gradients and the ice-bed interface is unlubricated; oblique ribbed bedforms develop beneath lateral shear margins, whereas transverse ribbed bedforms develop below frontal lobes. We infer that (i) ribbed bedforms strike orthogonal to the compressing axis of the horizontal strain ellipse of the ice surface and (ii) their development reveals distinctive types of subglacial drainage patterns: linked-cavities below lateral shear margins and efficient meltwater channels below frontal lobes. These ribbed bedforms may act as convenient geomorphic markers to reconstruct lateral and frontal margins, constrain ice flow dynamics and infer meltwater drainage characteristics of palaeo-ice streams.

## 1 Introduction

The dynamics of ice sheets is largely controlled by the activity of narrow corridors of fast-flowing ice, named ice streams (Paterson, 1994). In Antarctica, 10% of the ice sheet is covered by ice streams and 90% of the ice sheet is estimated to discharge through these corridors that flow at a few hundreds of $m \cdot y^{-1}$ (Bamber et al., 2000). Studies on modern ice streams – through remote sensing, borehole observations and geophysical measurements – have improved our understanding of the influence of ice streams on the mass balance of ice sheets. Many investigations highlight the role of lateral shear margins and frontal lobes on the overall stress balance of ice streams and the importance of imparted drag, along with basal drag, in controlling flow speed and ice stream width and length (Engelhardt et al., 1990; Goldstein et al., 1993; Echelmeyer et al., 1994; Patterson, 1997; Tulaczyk et al., 2000; Raymond et al., 2001; Kyrke-Smith et al., 2014; Minchew and Joughin, 2020). A lack of knowledge remains however on the spatial and temporal evolution of basal conditions beneath ice streams (e.g. distribution of basal shear stresses, meltwater drainage patterns) and on the processes acting at the ice-bed interface during the formation of subglacial bedforms. This is because basal investigations on modern ice streams are restricted to punctual observations. Palaeoglaciology aims to fill this gap by using the geomorphological and sedimentological records of former ice stream beds. Based on this approach, conceptual models of ice stream landsystems

characterized by either synchronous or time-transgressive imprints have been developed (Dyke and Morris, 1988; Kleman and Borgström, 1996; Clark and Stokes, 2003).

Even though general models of ice stream landsystems have now been established, the genesis of some landforms in these landsystems remains debated, notably their relation to ice dynamics and subglacial meltwater drainage. After mega-scale glacial lineations and drumlins, ribbed bedforms – subglacial periodic ridges that form transverse or oblique to the ice

flow direction – potentially represent some of the most conspicuous and ubiquitous landforms on palaeo-ice stream beds (Stokes, 2018). However, their distribution is not clearly accommodated in current ice stream landsystem models (Dyke et al., 1992; Clark and Stokes 2001; Stokes et al., 2007). Their formation is attributed either to (i) bed deformation (e.g. Boulton, 1987; Hättestrand and Kleman, 1999; Lindén et al., 2008), (ii) meltwater flow (e.g. Shaw, 2002), or (iii) a combination of these (e.g. Fowler and Chapwanya, 2014). The formation of ribbed bedforms by most of these mechanisms

involves spatio-temporal variations in basal shear stress. Such variations are observed across lateral shear margins (Raymond et al., 2001) and frontal lobe margins (Patterson, 1997) of ice streams. These margins should therefore constitute preferential areas for the formation of ribbed bedforms. However, a possible relation between ribbed bedforms and ice stream margins has not been reported so far.

In the following sections, we present a thorough review of the literature on shape, size, spatial distribution and formation

of ribbed bedforms and then we provide new constraints on the formation and distribution of ribbed bedforms beneath ice stream margins. For that purpose, we developed an experimental model that reproduces the dynamics of ice stream margins together with the evolution of subglacial bedforms and meltwater drainage. From remotely sensed images and topographic data, we also mapped ribbed bedforms below lateral shear and frontal lobe margins of palaeo-ice streams of the Laurentide Ice Sheet (North America). By combining the experimental and geomorphological results, we propose an

integrated model for the formation of ribbed bedforms associated with ice stream margins and discuss their palaeo-glaciological implications for the mapping of palaeo-ice streams and the reconstruction of past subglacial hydrology and ice stream dynamics.

## 2 Ribbed bedforms in ice stream landsystems

Glaciological, geomorphological and sedimentological studies on terrestrial- and marine-terminating ice streams have

enabled the development of landsystem models (**Fig. 1a**; Dyke and Morris, 1988; Kleman and Borgström, 1996; Clark and Stokes, 2003) which are critical for reconstructing spatial and temporal variations in the distribution of palaeo-ice streams, as well as understanding their geomorphic imprints across the landscape (Denton and Hughes, 1981; Patterson, 1997; Winsborrow et al., 2004; Margold et al., 2015). The areas where the ice starts to channel in the upstream part of ice streams (i.e. onset areas) are characterized by slow ice flow and a large spectrum of flow orientations due to the

convergence of multiple tributaries. Ice is then incorporated within narrow and well-defined trunks with high flow velocities (>300 m·y$^{-1}$; Clark and Stokes, 2003) and marked by swarms of streamlined bedforms parallel to ice flow direction (**Fig. 1**; mega-scale glacial lineations, flutes, drumlins and mega-grooves; Menzies, 1979; Stokes and Clark, 2001, 2002; Newton et al., 2018). Ice stream trunks are delimited by abrupt lateral shear margins, frequently underlined by continuous and linear shear moraines (Dyke and Morris, 1988; Dyke et al., 1992; Stokes and Clark, 2002b). Terminal

zones of ice streams are characterized either by marine-based ice shelves and calving fronts, or land-based marginal lobation (Stokes and Clark, 2001). At the margin of terrestrial terminating ice streams, belts of sediment ridges and mounds mimicking the lobe curvature can form in response to the ice lobe activity; stagnations produce hummocky moraines (e.g. Johnson and Clayton, 2003; Ebert and Kleman, 2004), advances produce push moraines, end moraines and overridden moraines (Totten,1969; Boulton, 1986; Evans and Hiemstra, 2005), and retreats produce recessional moraines

(Chandler et al., 2016) (**Fig. 1**).

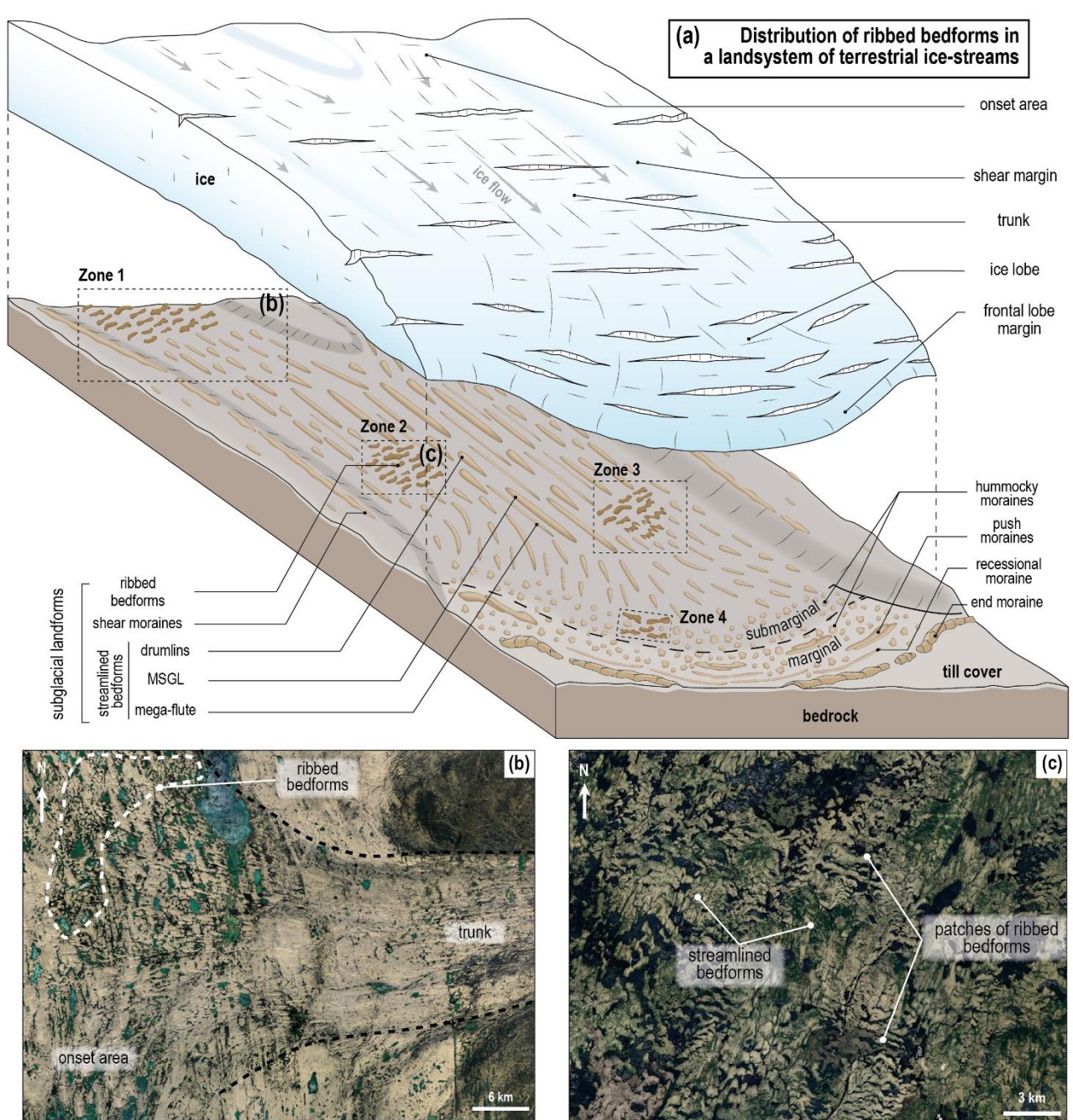

**Figure 1 (a) Distribution of ribbed bedforms, as currently envisaged, in an "ideal" isochronous landsystem of terrestrial palaeo-ice stream (modified after Clark and Stokes, 2003). (b) Ribbed bedforms located in the cold-based onset area (Zone 1), upstream of the streamlined bedforms recording the warm-based trunk of an ice stream (Fisher Lake, Prince of Wales Island, Canada). (c) Patches of ribbed bedforms with abrupt borders with surrounding streamlined bedforms (Zone 2; Lake Naococane, Canada). Also documented, and shown on a), but not shown with specific examples are progressive downstream transitions from ribbed bedforms to drumlins (Zone 3) and marginal relationships between hummocky and ribbed bedforms (Zone 4) (© Landsat / Copernicus - © Google Earth).**

Within this general model, the definition and genesis of ribbed bedforms remains unclear. Different types and shapes have been described, different theoretical models have been proposed for their formation, various interpretations exist for their timing and conditions of formation compared to ice and subglacial drainage dynamics, and their spatial distribution within glacial landsystems is still debated.

## 2.1 Types and shapes

Ribbed bedforms are subglacial ridges of sediment, periodic, and transverse or oblique to the palaeo-ice flow (**Fig. 1**). We use the term "ribbed bedform" as a descriptive morphologic term, without a genetic connotation, that embraces the following bedforms described in the literature: (i) Rogen moraines (Lundqvist, 1969), (ii) ribbed moraines (Hughes,

1964), (iii) mega-scale transverse bedforms (Greenwood and Kleman, 2010) and (iv) intermediate-sized bedforms interpreted as the topographic expression of traction ribs (Sergienko and Hindmarsh, 2013; Stokes et al., 2016).

Ribbed and Rogen moraines refer to sediment ridges originally described during the 1960s in North America and in Rogen Lake (Sweden) respectively. These bedforms exhibit a range of possible shapes and orientations, from transverse to oblique, relative to the palaeo-ice flow direction (Aylsworth and Shilts, 1989; Dunlop and Clark, 2006b). They are hundreds of metres to a few kilometres in length, a few hundreds of metres in width (l/w = 2.5), tens of metres in height, and their spacing is an order of magnitude lower than the ice thickness (Paterson, 1972; Dunlop and Clark, 2006b).

Mega-scale transverse bedforms are 20 to 40 km in length, 3 to 6 km in width (l/w = 5.7) and 5 to 10 m in height (Greenwood and Kleman, 2010). They are depicted with a perpendicular orientation relative to the ice flow direction and their spacing is an order of magnitude higher than the thickness of the overlying ice.

Intermediate-sized ribbed bedforms, interpreted as the morphological expression of traction ribs, are 1 to 6 km long, 0.4 to 2 km wide (l/w = 2.9) and 10 to 20 m high. They are believed to form perpendicular (90°) to oblique (20°) relative to the ice flow direction and below ice thicknesses equal to or an order of magnitude lower than their spacing (Sergienko and Hindmarsh, 2013; Stokes et al., 2016).

Crevasse-squeezed ridges also are subglacial periodic ridges oriented transversally or oblique to the ice flow (Evans et al., 2015; 2016). However, their shapes and metrics differ from those of the bedforms described above and are therefore not considered as ribbed bedforms here.

## 2.2 Spatial distribution in ice stream landsystems

The palaeo-glacial community agrees that some ribbed bedforms develop along rough ice-bed interfaces (frozen, unlubricated or associated to stiff beds) with high basal shear stresses under slow-flowing portions of ice sheets. Ribbed bedforms are thus believed to be ubiquitous from the inner, slow-moving, regions of ice sheets (Aylsworth and Shilts, 1989; Dyke et al., 1992; Hättestrand and Kleman, 1999; Greenwood and Kleman, 2010; Stokes, 2018) up to the onset areas of ice streams at the transition between cold- and warm-based ice, where the ice flow velocity increases downstream (**Fig. 1b - Zone 1**; Bouchard, 1989; Dyke et al., 1992; Hättestrand and Kleman, 1999). They are thought to be less common in ice stream trunks, where they would occur only as isolated patches within a mosaic of sticky and slippery spots associated with streamlined bedforms (**Fig. 1c - Zones 2, 3**; Cowan, 1968; Aylsworth and Shilts, 1989; Bouchard, 1989; Dyke et al., 1992; Stokes and Clark, 2003, Stokes et al., 2006a, 2008) and in submarginal areas, where they would be associated with hummocky moraines within stagnant ablation complexes (**Fig. 1 - Zone 4**; Marich et al., 2005; Möller, 2006, 2010). To summarize, within an isochronous glacial landsystem, ribbed bedforms are believed to form below large-scale portions of slow-flowing ice (ice domes and stagnant margins of ice sheets) and below localized patches of slow ice within fast-flowing ice streams (sticky spots). According to this model, they would develop contemporarily to streamlined bedforms, either upstream (from ice domes to onset areas of ice streams), downstream (along marginal ablation complexes) or laterally (within sticky spots).

## 2.3 Theoretical models of formation

Theoretical models of formation invoke either basal thermal conditions, meltwater flows or initial bed topography, to explain the shape and periodicity of ribbed bedforms. Four categories of formation processes have been proposed: (i) deformation or reshaping of pre-existing sedimentary mounds, such as former streamlined or marginal landforms, by overriding ice (Boulton, 1987; Lundqvist, 1989; Möller, 2006); (ii) fracturing and extension of frozen beds along transitions from warm-to-cold ice bases, where tensional stresses increase (Hättestrand and Kleman, 1999; Sarala, 2006); (iii) subglacial meltwater floods responsible for the formation of inverted erosional marks at the ice base, infilled by sediments (Shaw, 2002); (iv) till deformation, in response to the flow of ice over bed heterogeneities resulting from

variations in pore water pressure, basal thermal regime and bed strength (Terzaghi, 1931; Shaw, 1979; Bouchard, 1989; Kamb, 1991; Tulaczyk et al., 2000b; Lindén et al., 2008; Stokes et al., 2008). The last process is consistent with physically-based mathematical models demonstrating that ribbed bedforms naturally arise from wavy instabilities in the combined flow of ice, basal meltwater and viscoplastic till (Hindmarsh, 1998a,b; Fowler, 2000; Schoof, 2007; Clark, 2010; Chapwanya et al., 2011; Sergienko and Hindmarsh, 2013; Fowler and Chapwanya, 2014; Fannon et al., 2017). In conclusion, ribbed bedforms occur in a highly varied distribution within ice sheets and ice streams and multiple hypotheses have been proposed for their formation. A size and shape continuum has been suggested between ribbed, hummocky and streamlined bedforms, suggesting that these subglacial bedforms could form in response to the same governing processes modulated by either ice flow velocity or ice flow duration (Aario, 1977; Rose, 1987; Dunlop and Clark, 2006b; Stokes et al., 2013a; Ely et al., 2016; Fannon et al., 2017). Moreover, ribbed bedforms are frequently overprinted by drumlins and embedded within polygenetic landsystems, corresponding to multiphase stories, which complicate their interpretation (Cowan, 1968; Aylsworth and Shilts, 1989; Stokes et al., 2008). Previous studies emphasize the importance of ribbed bedforms in the reconstruction of basal shear stresses, ice-bed interactions, basal processes and ice dynamics (Dyke and Morris, 1988; Alley, 1993; Stokes et al., 2008; Stokes et al., 2016).

## 3 Methods

### 3.1 Analogue modelling

The development of subglacial bedforms potentially involves four mass transfer processes – erosion, transport, deposition and deformation – that can occur simultaneously and interact beneath ice streams, and four components - ice, water, till, and bedrock – that have complex and distinct rheological behaviours (Paterson, 1994). To understand the formation of ribbed bedforms, numerical models have been developed using naturally-occurring typical values for the physical properties of these components. In these models the growth of subglacial bedforms is based on (i) the development of along-flow instabilities which creates wave-like bedforms resulting from infinitesimal perturbations at the surface of a deformable till layer (Hindmarsh, 1998a,b; Fowler, 2000; Schoof, 2007; Chapwanya et al., 2011; Fowler and Chapwanya, 2014; Fannon et al., 2017) and (ii) till entrainment and deposition resulting from till flux (driven by bed deformation and ice pressure gradient) and lee-side cavity deposition (Barchyn et al., 2017). Three-dimensional numerical modelling of bedforms remains a challenging enterprise. Their investigation is far from complete, mostly because the involved components show drastically distinct thermo-dependent and strain-rate dependent rheologies, and therefore temporal scales of activity (Paterson, 1994). To circumvent the challenge of numerically modelling these complex interactions, Lelandais et al. (2016; 2018) developed a laboratory-based model capable of simultaneously simulating ice flow, subglacial hydrology and subglacial erosion/transport/sedimentation/deformation. This model contributed to better constrain the link between subglacial meltwater drainage and the lifecycle of ice streams, but subglacial bedforms were not included. We used the same approach with a new experimental setup, specifically designed to simulate the development of subglacial bedforms.

### 3.1.1 Experimental setup

The experimental device consists of a stainless steel box, 5 cm high, 2 m × 2 m wide, and surrounded by a gutter 5 cm in width (**Fig. 2a**). A 5 cm thick bed, composed of saturated fine sand (median grain size $d_{med}$ = 100 µm, density $\rho_{bulk}$ = 2000 kg·m$^{-3}$, porosity $\Phi_s$ = 41 %, permeability $K_s$ = 10$^{-4}$ m·s$^{-1}$) fills the box to simulate a soft, porous, permeable and erodible subglacial bed (**Fig. 2b**). Grains of coloured coarse sand (grain size d = 850 to 1250 µm), representing bedrock blocks, are sprinkled over the flat surface of the bed. To simulate the ice sheet, a circular cap of viscous and transparent silicone putty (density $\rho_{sil}$ = 967 kg·m$^{-3}$, viscosity $\eta_{sil}$ = 5.10$^4$ Pa·s$^{-1}$), 3 cm thick in its centre and 90 cm in diameter, covers the bed.

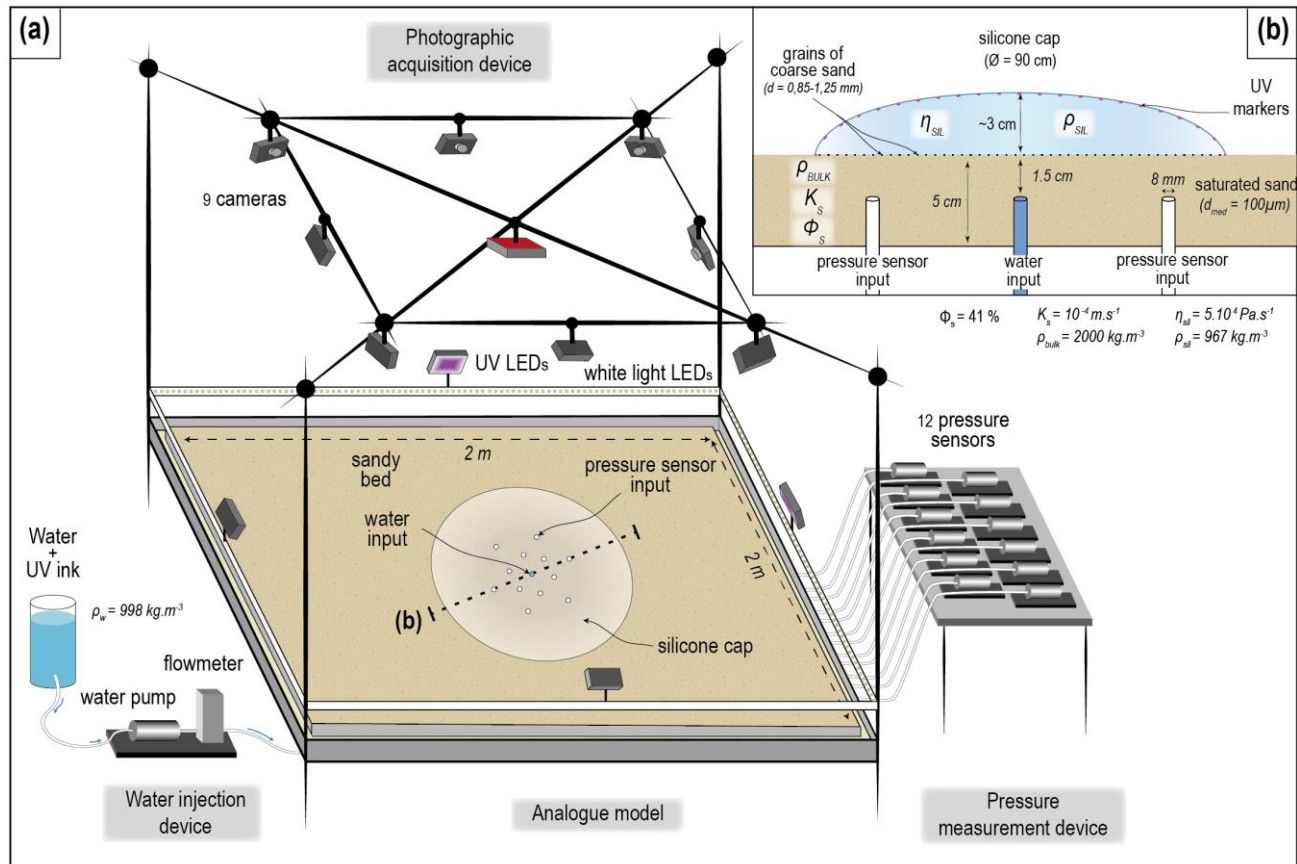

**Figure 2 (a) Experimental model and monitoring apparatus. A cap, of transparent and viscous silicone putty, flows under its own weight above a bed composed of water-saturated sand. We inject coloured water in the bed, through an injector located below the centre of the silicone cap. We use a photographic acquisition device and pressure sensors to monitor the development of bedforms at the silicone/bed interface, the silicone flow velocity, the water flow and the water pressure. (b) Transverse section of the model with the experimental parameters.**

We inject a solution of mixed water and UV ink (bulk density $\rho_w = 998$ kg·m$^{-3}$) through an injector located below the centre of the silicone cap in order to produce water flow beneath the silicone and highlight the flow pattern using UV light. The central injector, with a diameter of 8 mm, is placed 1.5 cm beneath the surface of the bed and connected to a water pump. The injection is regulated by a flow meter (discharge $Q = 0\text{-}50$ ml·min$^{-1}$) and is calculated to allow water circulation within the bed and along the silicone-bed interface, with a pressure of injected water exceeding the combined weights of the bed and silicone layers. Fifteen experiments, 60 min in duration, were performed with different injection scenarios: constant, binary and discontinuous (**Table 1**). Experimental scenario were replicated at least 3 times in order to ensure the reproducibility of the experimental set up. The experimental device is scaled according to the principles described in Lelandais et al. (2016; 2018): to take into account the intimate links between glacial dynamics, subglacial hydrology and subglacial landform development, all physical quantities in the experiment are defined so that the dimensionless ratio between ice velocity and incision rate of subglacial erosional landforms has similar values in the model and in nature. Lelandais et al. (2016; 2018) demonstrated that models based on this scaling ratio are able to reproduce landforms representative of subglacial systems.

### 3.1.2 Monitoring of experiments and post-processing of data

The experimental device is equipped with white light LEDs and UV LEDs, alternating every 15 s during an experiment. An array of nine cameras acquires photographs (every 15 s) both in white light (all cameras) and in UV light (red camera only; **Fig. 2a**). The transparency of the injected water and the silicone cap in white light enables manual mapping of bedforms and reconstruction of Digital Elevation Models (DEMs) of the bed surface from the white light photographs, with a precision of $\pm 0.1$ mm, through a photogrammetric method developed by Lelandais et al. (2016). The morphometric

properties of bedforms and drainage features (**Fig. S1**) are calculated from interpreted snapshots and DEMs. Maps of water distribution at the silicone-bed interface are derived from the UV photographs, thanks to the fluorescence of the injected water. An automatic treatment of these images and a calibration with DEMs of the sub-silicone bed allows fluorescence intensity to be converted into water thickness. The pore-water pressure is punctually measured, 1.5 cm beneath the bed surface, with twelve pressure sensors (8 mm in diameter) distributed in two concentric circles, 15 and 30 cm in radius respectively and centered on the central water injector (**Figs. 2a**, **b**). The horizontal positions of UV markers, 1 mm in diameter and placed with an initial spacing of 5 cm on the silicone surface, are monitored with a time step of 90 s (**Fig. 2b**). Horizontal velocity ($V_{surf}$) and deformation maps for the surface of the silicone cap are then calculated and interpolated from the temporal record of the marker displacements. The horizontal deformation of the silicone cap surface is quantified with two indicators, (i) the strain rate and orientation of the principal axes of the instantaneous strain ellipse ($\varepsilon_{extending} > 0$ and $\varepsilon_{compressing} < 0$; Ramsay and Huber, 1987), and (ii) the absolute magnitude of the horizontal shear strain rate ($\varepsilon_{shear}$; Nye, 1959). Those indicators are computed for each triangle of a mesh, established by a Delaunay triangulation of all the UV markers (**Fig. S2**).

**Table 1 Water injection scenarios and type of water drainage observed in each experiment.**

| Experiment number | Water injection scenario | Channelized drainage beneath lobes | |
|:---:|:---:|:---:|:---:|
| | | **Well-developed** | **Poorly-developed** |
| 1 | | ■ | |
| 2 | Constant | ■ | |
| 3 | 25 ml·min$^{-1}$ (60 min) | ■ | |
| 4 | | ■ | |
| 5 | | ■ | |
| 6 | | ■ | |
| 7 | Constant | | ■ |
| 8 | 37.5 ml·min$^{-1}$ (60 min) | ■ | |
| 9 | | | ■ |
| 10 | Binary | ■ | |
| 11 | 25 ml·min$^{-1}$ (30 min) - 50 ml·min$^{-1}$ (30 min) | ■ | |
| 12 | | ■ | |
| 13 | Discontinuous | ■ | |
| 14 | 12.5 ml·min$^{-1}$ (15 min) - 25 ml·min$^{-1}$ (15 min) - 0 ml·min$^{-1}$ (15 min) - | | ■ |
| 15 | 25 ml·min$^{-1}$ (5 min) - 37.5 ml·min$^{-1}$ (5 min) - 50 ml·min$^{-1}$ (5 min) | ■ | |

### 3.1.3 Physics of silicone-water-bed interactions in the experiment

The model is designed to explore the basic mechanical interactions between a simplified water-routing system, a deformable and erodible sedimentary bed and an impermeable viscous cover. The formation of sub-silicone bedforms in the experiments involves interactions between the silicone putty, the injected water and the sand bed. Compared with ice, the silicone putty is Newtonian, isotropic and impermeable. Under the experimental conditions (between 15-20°C and at atmospheric pressure), its viscosity is nearly independent of temperature, and the bed is constantly wet and saturated. Consequently, temperature-dependent processes (shear heating, heat softening, melting, and freezing) and shear softening or shear hardening related to the non-Newtonian behaviour or the anisotropy of ice are not reproducible. Unlike ice, the Newtonian silicone putty is thus unable to localise viscous deformation when stress increases (in particular along lateral shear margins) or to produce fractures in the range of experimental flow velocities we can simulate. Consequently, spatial velocity gradients are expected to be smaller in the experiment than in nature, and the width of experimental lateral shear

margins is overestimated compared to the width of experimental ice streams. The use of silicone putty also induces a

major scaling limitation since the viscosity ratios between the cap materials, either ice or silicone putty, and the basal fluid, water in both cases, are different. The size of erosive channelized features produced in the analogue model are thus overestimated compared to the dimensions of experimental ice streams. Internal-production of meltwater, complex spatial variations of subglacial hydrology and lobe margin ablation and retreat are not reproducible either. The water-saturated sand bed is homogeneous and can deform by both localised and diffuse intergranular shearing; the flow of both water and

silicone can thus produce both internal deformation of the bed and erosion, transport and deposition of grains at the bed/water or bed/silicone interface. The transport of sand grains through supra-silicone entrainment and incorporation into basal silicone are not simulated.

The model, like all models, is not perfectly realistic since it reproduces neither a complete and scaled miniaturization of nature nor all subglacial physical processes. Previous physical experiments reproduced subglacial channelized features

(tunnel valleys) and marginal landforms (ice-contact fans and outburst fans), and suggested a close relationship between the development of tunnel valleys and the stabilization of analogue ice streams (Lelandais et al., 2016; 2018). It turns out that experimental model, although imperfectly scaled, reproduces landforms visually conforming to subglacial landforms, emphasizing the "unreasonable effectiveness" of analogue modelling (Paola et al., 2009).

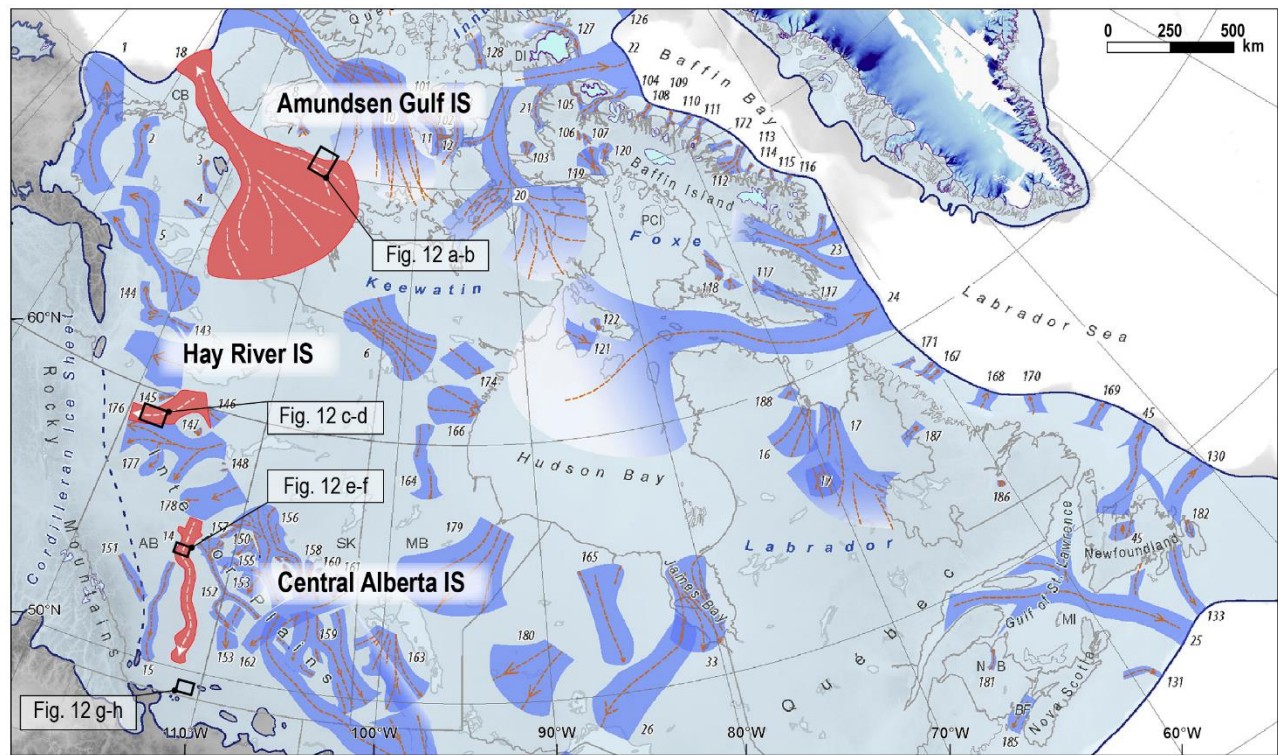

**Figure 3 Extent of the North American Laurentide Ice Sheet during the Last Glacial Maximum with positions of known ice stream tracks (modified from Margold et al., 2018). The areas mapped in this study in three ice streams (Amundsen Gulf, Hay River and Central Alberta IS) are identified by black rectangles.**

### 3.2 Mapping and morphometric analysis on palaeo-ice stream landsystems

To identify, map and characterize ribbed bedforms associated with natural palaeo-ice stream margins, we compiled

DEMs, hillshades and ice stream contours of the North American Laurentide Ice Sheet (Margold et al., 2015) in a Geographic Information System (GIS). Four distinct regions (**Fig. 3**), covering a few thousand square kilometres, overlapping the lateral shear and frontal lobe margins of three palaeo-ice stream beds were studied: (i) the Amundsen Gulf Ice Stream (21.8 – 12.9 cal ka BP), (ii) the Central Alberta Ice Stream (20.5 – 17 cal ka BP) and (iii) the Hay River Ice Stream (13.9 – 12.9 cal ka BP; Margold et al., 2018). We used a 15 m LiDAR bare-earth DEM supplied by the Alberta

Geological Survey for the Central Alberta and Hay River ice streams, and a 10 m Digital Surface Model computed by

optical stereo imagery (ArcticDEM; Porter et al., 2018) for the Amundsen Gulf Ice Stream. By means of break in slopes observed on hillshades, we digitized ribbed bedforms, streamlined bedforms indicative of predominant ice flow directions and other landforms indicative of ice stream lateral and frontal margins at the scale 1:50 000. The length (long-axis), width (short-axis) and orientation of ribbed bedforms compared to ice flow direction were extracted using the 'Minimum

Bounding Geometry' tool in ArcGIS. The orientation values of ribbed bedforms and streamlined bedforms were compiled in rose diagrams for each selected region.

## 4 Results

### 4.1 Stream dynamics and development of ribbed bedforms in the experiment

Without obvious dependence on the injection scenarios, the experiments listed in **Table 1** produced two types of streams

and lobes related to two different types of water drainage networks beneath the silicone cap: (i) well-developed channelized drainage (n = 12 experiments) and (ii) poorly-developed channelized drainage (n = 3 experiments). In the following section, we present the evolution of silicone stream dynamics and ribbed bedforms for one experiment representative of each kind: experiments 10 and 14, respectively (**Table 1**). In those two experiments, the water injection discharge was doubled in stage 3 (Q = 50 ml·min⁻¹) compared to stages 1 and 2 (Q = 25 ml·min⁻¹).

In both experiments, when water injection starts, a circular water pocket forms and grows along the silicone-bed interface until it reaches a diameter of 15 to 25 cm. This pocket migrates towards the margin of the silicone cap, as a continuous water film, until it suddenly drains outside the cap. The presence and migration of the water pocket induces a local increase in silicone flow velocity (from $V_{surf} = 0.2 \times 10^{-2}$ mm·s⁻¹ to $V_{surf} = 10 \times 10^{-2}$ mm·s⁻¹) forming a 20 cm wide stream that propagates from the margin towards the centre of the silicone cap. A lobe forms at the extremity of the fast-flowing

corridor in response to this drainage and surge event. In Experiment 10, seven water pockets successively form and produce an equal number of drainage events and lobe advances; in Experiment 14, only one drainage event occurs (**Fig. 4**). The drainage events generate marginal deltas along the silicone margin, constituted by sand grains eroded from the bed during the migration of the water pocket. The sustainability of a water film at the silicone-bed interface, once the initial drainage event is over, contributes to maintain a fast and durable silicone flow, the silicone stream switches on and

the silicone lobe keeps growing.

Landforms produced in the experiments are illustrated and annotated in **Fig. 5**: these include grooves, ribbed and hummocky bedforms beneath the streams, and marginal deltas and pushed sediments in front of the lobes. The successive stages (**Figs. 6 to 9**) are described below in relation to: (i) the stream evolution derived from surface measurements, (ii) the formation, evolution and morphometric characteristics of bedforms derived from photographs in white light and

DEMs, and (iii) the evolution of water drainage systems derived from UV photographs and DEMs.

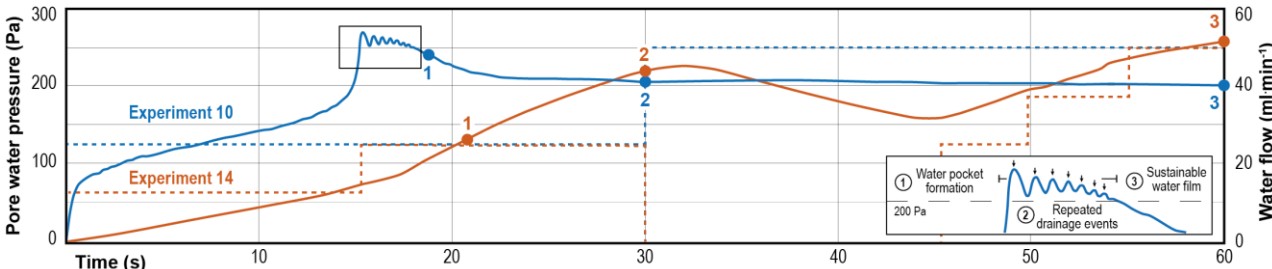

**Figure 4** Evolution of pore water pressure (solid lines) measured in the trunk axis (see Fig. 5a for position) and water flow injected in the bed through the water injector (dotted lines), for Experiments 10 and 14. Points numbered 1 to 3 correspond to stages described in sections 4.1, 4.2 and 4.3.

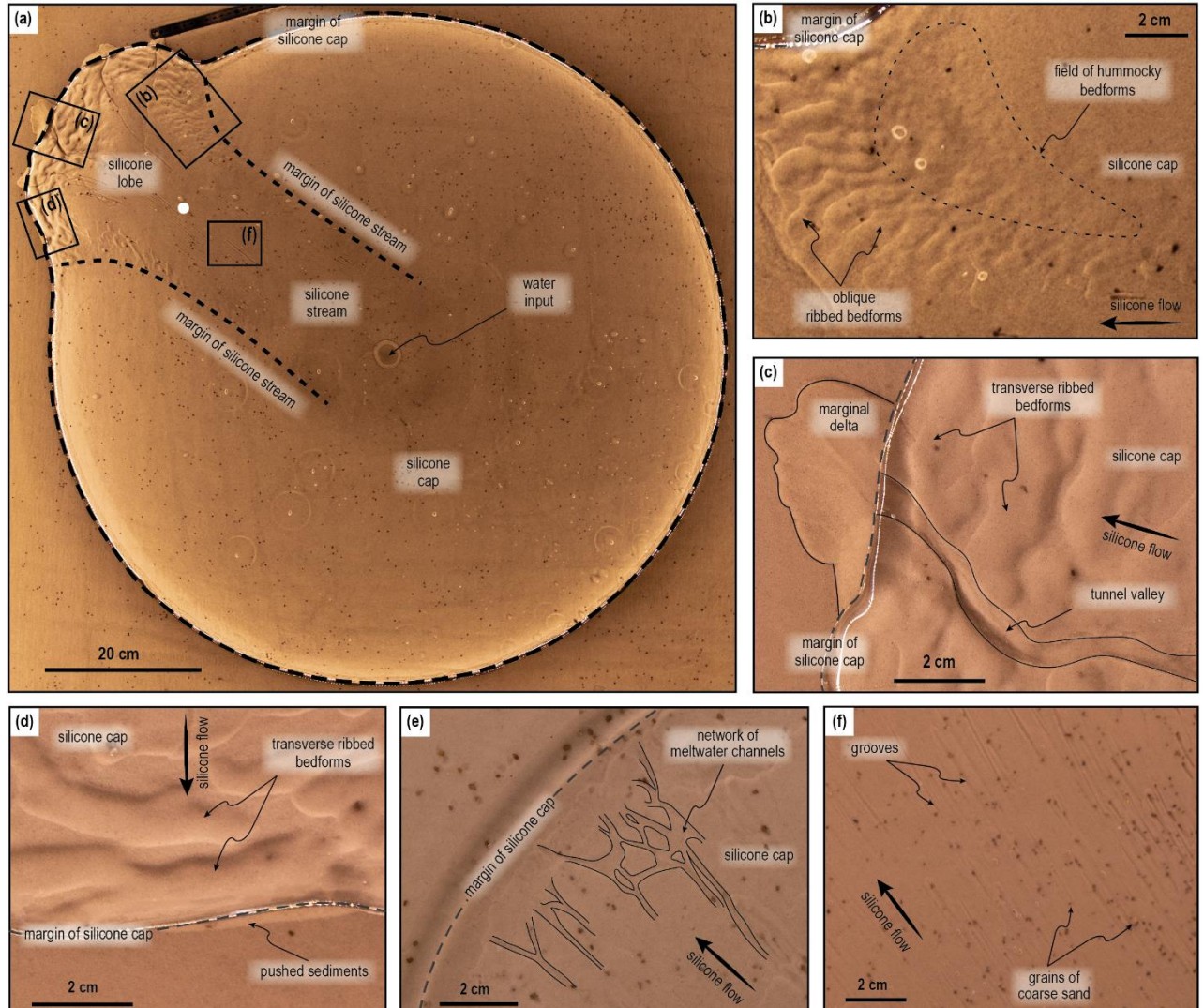

**Figure 5 Illustration of bedform assemblages produced during the experiments. (a) Overall surface view of the silicone cap at the end of an experimental run, showing the distribution of bedforms beneath a stream/lobe system characterized by a well-developed channelized drainage (Experiment 10). The white dot indicates the position of the pressure sensor where pore water pressure presented in Fig. 4 is measured. (b) Close-up on oblique ribbed and hummocky bedforms observed at the right lateral shear margin. (c) Transverse ribbed bedforms parallel to the lobe frontal margin and formed submarginally. Tunnel valley and associated marginal delta. (d) Transverse ribbed bedforms parallel to the lobe frontal margin and pushed marginal sediments bulldozed by the lobe margin. (e) Network of meltwater channels extending parallel and perpendicular to the silicone flow (not located on (a), because it was observed in Experiment 14, characterized by a poorly-developed channelized drainage). (f) Field of grooves inscribed in the bed and oriented parallel to silicone flow with embedded sand grains at their down-stream extremities.**

### 4.1.1 Stage a - Initiation of ribbed bedforms beneath incipient shear and lobe margins

The streams flow 10 to 20 times faster (**Fig. 7a**: $V_{surf} = 0.08$ mm·s$^{-1}$; **Fig. 9a**: $V_{surf} = 0.17$ mm·s$^{-1}$) than the inter-stream areas ($V_{surf} < 0.01$ mm·s$^{-1}$), inducing velocity gradients across the lateral margins of the streams. The compressing strain rate of the strain ellipse and the shear strain rate are zero in the centre of the stream trunk and in the surrounding ice cap, while they are maximal along lateral margins of the streams where the velocity gradient is the highest (**Figs. 7a, 9a**; $\varepsilon_{compressing} = -2.5 \times 10^{-4}$ s$^{-1}$; $\varepsilon_{shear} = 3$ to $5 \times 10^{-4}$ s$^{-1}$). These patterns of high shear deformation define symmetrical bands on the silicone surface, and resemble those of natural lateral shear margins (Echelmeyer et al., 1994; Raymond et al., 2001). In experiment 10 (**Fig. 7a**), the stream velocity decreases from the centre ($V_{surf} = 0.08$ mm·s$^{-1}$) toward the frontal and lateral margins (respectively, $V_{surf} = 0.06$ mm·s$^{-1}$ and $V_{surf} = 0.04$ mm·s$^{-1}$), while the silicone-bed interfaces of these margins are dewatered (**Fig. 7a**; water thickness = 0). Beneath the lobe, meltwater channels, 50 to 70 mm in length and 4 to 8 mm in width, start to develop by regressive erosion from the margin and initiate channelization of the water flow (**Figs. 6a, 8a**). Sand grains washed away by bed erosion are transported within these incipient channels towards the marginal deltas

(**Figs. 5c, 6a**). Radial growth and advance of the lobe over the marginal deltas form ridges of pushed sediments, 5 mm in width and 40 mm in length (**Figs. 5d, 6a**). Below the streams (**Fig. 6a, 8a**), some coarse sand grains are lodged in the base of the fast-flowing silicone and carve the bed to form sets of parallel grooves (1 mm wide, 10 to 20 mm long) highlighting the stream trunk and the direction of the silicone flow (**Figs. 5f**). Below frontal and lateral margins, fields of periodic ridges develop transverse to ice flow, with undulating crests and an average wavelength of 8 mm (**Figs. 6a, 8a**). They are on average a few mm thick, 7 mm wide and 30 mm long, i.e. 4 times larger in length than in width. Their shapes (undulating crest, elongation ratio) and periodic spatial organization (**Figs. 5b, d**) resemble those of ribbed bedforms observed in glacial landsystems. In the experiments, ribbed bedforms initiate perpendicular (90°) to oblique (60°) to the compressing axis of the instantaneous strain ellipse measured at the surface of the overlying silicone (**Fig. 7a, 9a**). Most of them form below the lobe margins perpendicular to the silicone flow direction, and some scattered and isolated ones form below the lateral margins perpendicular (90°) to oblique (45°) to the silicone flow. Other bedforms with circular to ovoid shapes (mounds) form in between the ribbed bedforms. These mounds are typically less than 0.1 mm high, 4 mm wide and 6 mm long, with an average spacing of 10 mm. Their long-axes do not show a preferential orientation (**Figs. 6a, 8a**). The shape and spatial organisation of these mounds (**Fig. 5b**) resemble those of some hummocky bedforms in glacial landsystems.

### 4.1.2 Stage b – Channelization of the drainage system below protruding lobes and evolution of ribbed bedforms

After 30 min, channelized drainage networks develop beneath the silicone lobe, draining water fed from the upstream water film towards the margin. The morphological evolution of these networks differs between both types of experiments, and we suggest is a controlling influence on the distinctive evolution of stream dynamics from this point. From the previous stage, the incipient network of meltwater channels in Experiment 10 evolves into tunnel valleys – characterized by undulating long profiles with over-deepening and adverse slopes - (**Fig. 5c**), increasing in width (up to 1.2 cm), length (15 cm) and depth (0.07 cm), and marginal deltas continue to grow (**Fig. 6b**). Three individual tunnel valley systems, two slightly sinuous valleys with some tributaries and one anastomosed valley network, are observed (**Fig 6b**). Simultaneously, the stream velocity decreases by half, resulting in a slowdown in the advance of the frontal lobe margin (**Fig. 6b**: 28 cm in width; **Fig. 7b**: $V_{surf} = 0.04$ mm·s$^{-1}$). Proportionally to the slowdown intensity, the lateral shear margins record a halving of shear strain rate (**Fig. 7b**: $\varepsilon_{shear} = 1.25 \times 10^{-4}$ s$^{-1}$) and strain ellipse deformation ($\varepsilon_{extending} = + 2.5 \times 10^{-4}$ s$^{-1}$; $\varepsilon_{compressing} = - 1.3 \times 10^{-4}$ s$^{-1}$).

In contrast to the above, in Experiment 14 the pre-existing meltwater channels remain shallow and keep a constant width, but lengthen downstream in response to the migration of the lobe margin (**Fig. 8b**: 0.4 cm wide, 12 cm long, 0.04 cm deep). Meltwater channels are rectilinear with poorly developed tributaries and lack deltas at their downstream extremities (**Fig. 5e**). In parallel, the silicone stream flow maintains a high velocity, although a small deceleration occurs (**Fig. 9b**: $V_{surf} = 0.13$ mm·s$^{-1}$), sustaining lobe growth (36 cm in width). As a result, the lateral shear margins record a small decrease of the shear strain rate (**Fig. 9b**: $\varepsilon_{shear} = 3.5 \times 10^{-4}$ s$^{-1}$) and a downstream-to-upstream decrease of the strain ellipse deformation ($\varepsilon_{extending} = + 4.4 \times 10^{-4}$ s$^{-1}$; $\varepsilon_{compressing} = - 1.9 \times 10^{-4}$ s$^{-1}$).

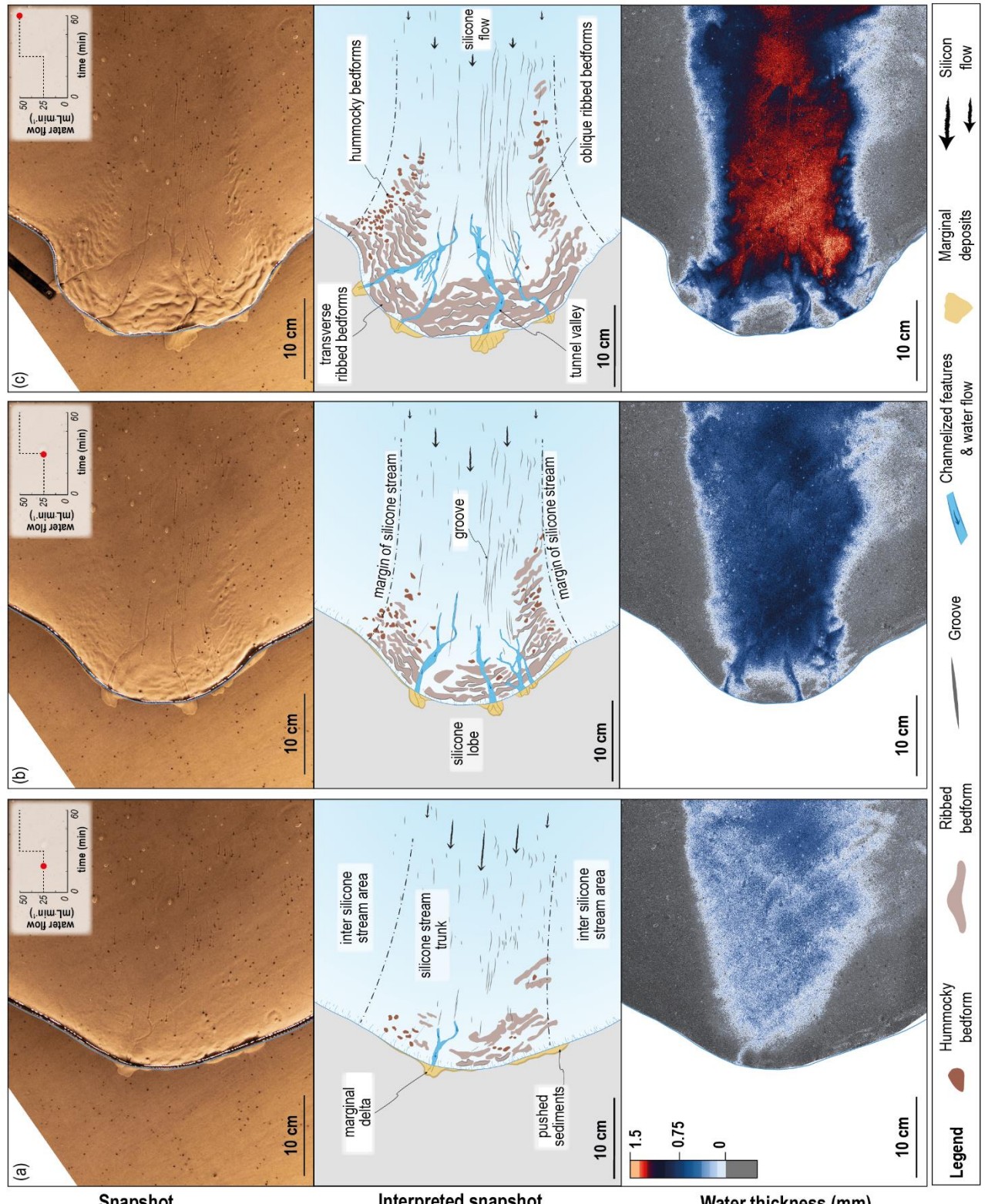

**Figure 6 Temporal evolution (stages a, b and c) of the silicone stream and lobe system, for a typical experiment with a well-developed channelized drainage system (Experiment 10), illustrated with snapshots representative of the three main stages of development. The upper, intermediate and lower panels show, respectively, (i) white-light photographs of the surface of the model, (ii) interpretations of bedforms observed on the photographs and (iii) thickness (in mm) of the water film flowing beneath the silicone cap, derived from UV photographs.**

350

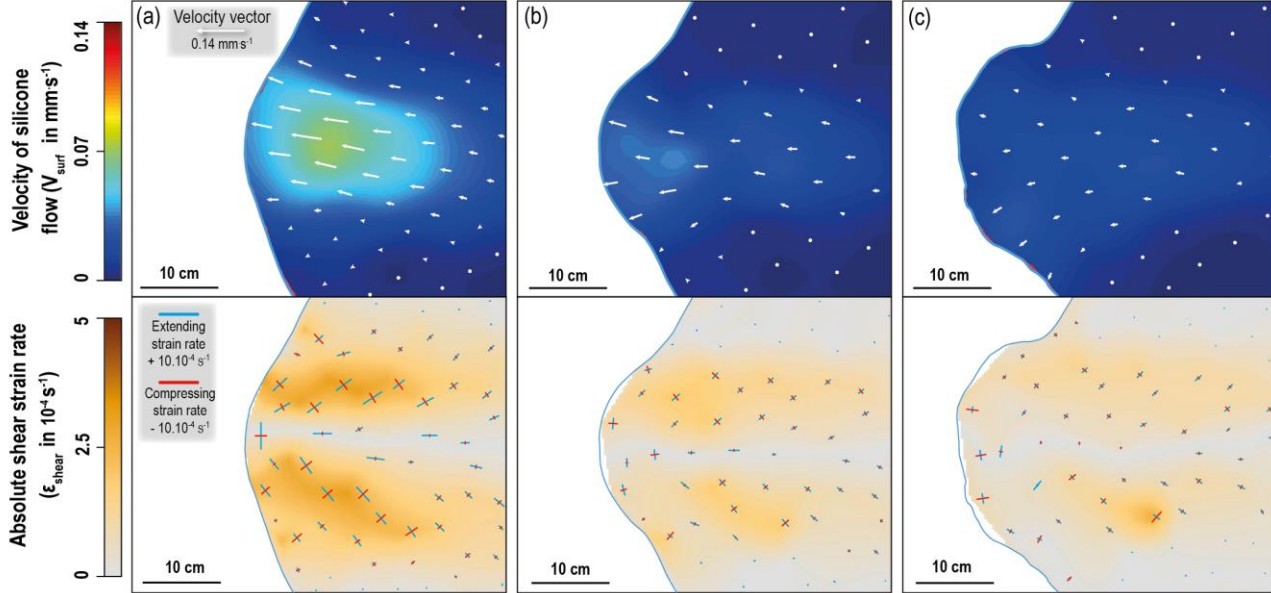

**Figure 7** Temporal evolution (same stages a, b and c as Fig. 6) of the silicone stream and lobe system, for a typical experiment with a well-developed channelized drainage (Experiment 10). The upper and lower panels show respectively, (i) the velocity field of silicone flow (mm·s⁻¹) and the velocity vector maps of the silicone cap upper surface and (ii) extrapolated maps of absolute shear strain rate (s⁻¹) and the orientation of the principal axes of the instantaneous strain ellipse whose length correspond either extending or compressing strain rate (ε in s⁻¹),. Maps highlight the development of the fast-flowing corridor (i.e. silicone stream) surrounded by stagnant or very slow-moving silicone (i.e. inter stream area) and the formation of two symmetrical shear bands on both sides of the stream (i.e. the lateral shear margins).

Within the trunk of both experiments, the orientation and direction of the grooves matches the velocity vectors of the silicone flow. The grooves are restricted to the fast-flowing corridor and display lengths up to 5 cm and 8 cm for Experiments 10 and 14 respectively, and they tend to keep a constant width of approximately 0.1 cm. Grooves remain rectilinear and parallel to mean silicone displacement in the upstream part of the streams but diverge and curve in the downstream area following the fan-shaped pattern of silicone velocity vectors in the lobe (**Figs. 6b, 8b**). During this second stage, the ribbed bedforms keep growing but differ below lateral shear margins (i.e. lateral ribbed bedforms) and frontal lobes (i.e. submarginal ribbed bedforms). Below shear margins, new lateral ribbed bedforms with arcuate planforms progressively develop oblique (45 to 60°) to silicone velocity vectors and perpendicular to compressing axes of the horizontal strain ellipses. They first appear near the lobe and then develop toward the upstream area of the trunk borders. Pre-existing and newly formed lateral ribbed bedforms increase in length and width, parallel to the extending axes and perpendicular to the compressing axes of the horizontal strain ellipse respectively (**Figs. 6b, 8b**; 1 cm wide and 5 cm long). Their wavelength is typically 0.6 to 0.7 cm in Experiment 10 (**Fig. 6b**), while it is 1.0 to 1.5 cm in Experiment 14 (**Fig. 8b**). Beneath the lobe margin of Experiment 10, the water film ceases to exist in-between the tunnel valleys (**Fig. 6b**). These water-free areas coincide with the growth of pre-existing submarginal ribbed bedforms and the formation of new ones parallel to the lobe margin and perpendicular to silicone velocity vectors and compressing axes (**Fig. 7b**). They develop below the lobe from (i) the initial sandy bed and (ii) the recycled marginal landforms (i.e. marginal deltas and pushed sediments). The development and the coalescence of these bedforms below the lobe tend to form a belt composed of broad, arcuate and transverse submarginal ribs (**Fig. 6b**; up to 1.6 cm in width, up to 8 cm in length and 1.1 cm in wavelength). In Experiment 14, characterized by smaller meltwater channels, only few and scattered submarginal ribbed bedforms – most are inherited from the previous stage and a few newly formed – of smaller dimensions appear (**Fig. 8b**; up to 0.8 cm in width and up to 4 cm in length). The population of mounds resembling hummocky bedforms keeps increasing (from 15 to 40 in Experiment 10, and from 0 to 17 in Experiment 14). Some mounds observed in the previous stage have evolved into ribbed bedforms, and new mounds form with constant dimensions (0.5 cm in width, 0.7 cm in length), especially in the bending zone between lobe borders and shear margins (**Figs. 6b, 8b**).

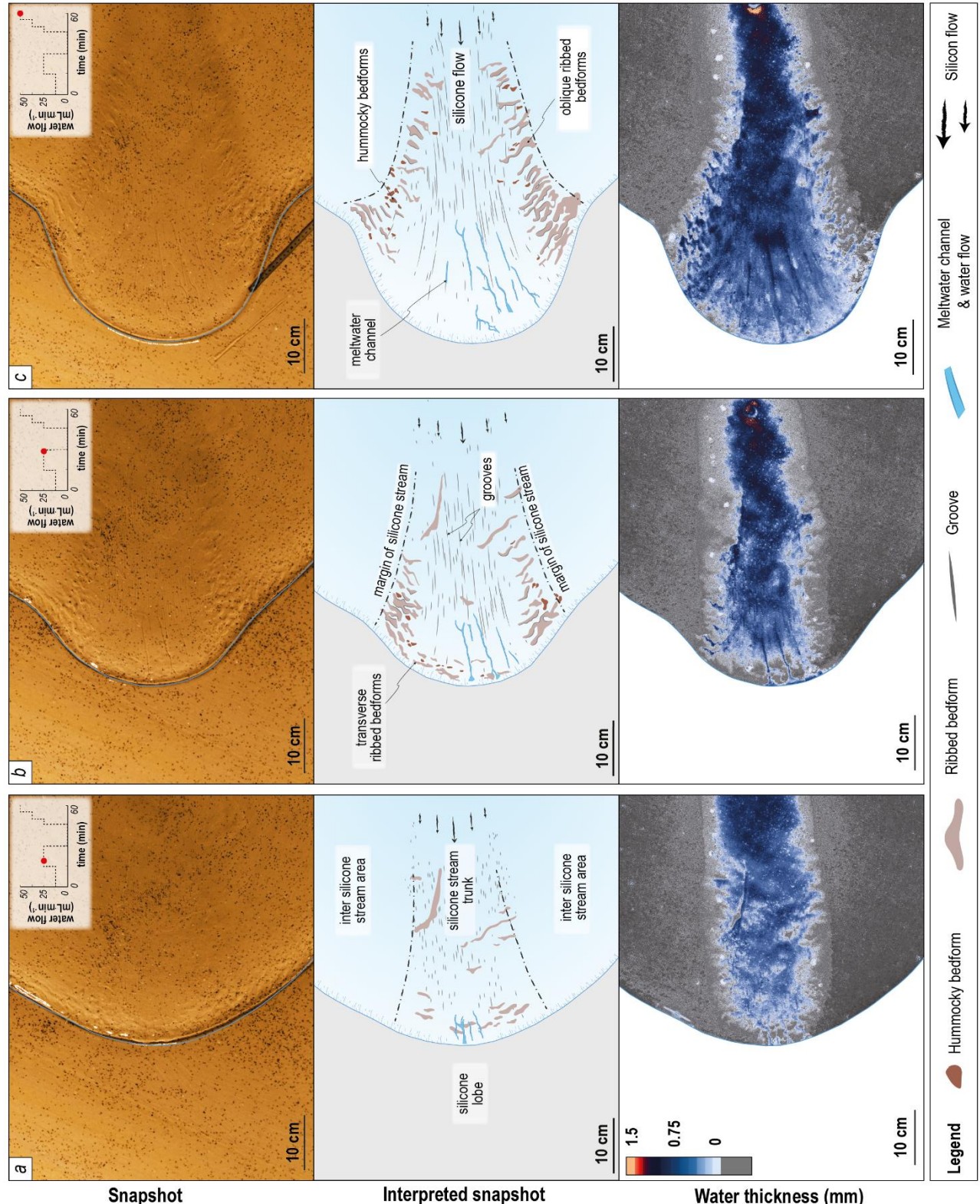

385

**Figure 8 Temporal evolution (stages a, b and c) of the silicone stream and lobe system, for a typical experiment with a poorly-developed channelized drainage (Experiment 14). Same panels as Figure 6.**

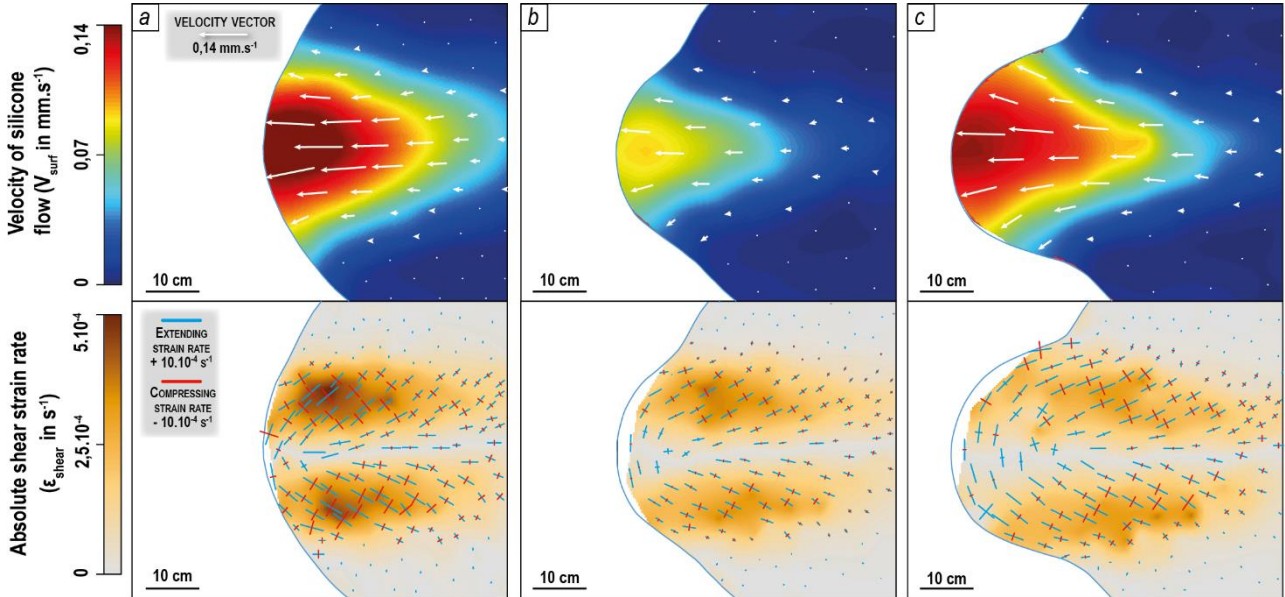

**Figure 9 Temporal evolution (same stages a, b and c as Fig. 8) of the surface behaviour of the silicone stream and lobe system, for a typical experiment with a poorly-developed channelized drainage (Experiment 14). Same panels as Figure 7.**

### 4.1.3 Stage c – Morphological response to changes in drainage characteristics

After 60 min, the two experimental runs described here show distinctive drainage systems (well-developed vs. poorly-developed), silicone streams dynamics and landform development histories.

**Well-developed channelized drainage system (Experiment 10)**

In response to the imposed increase in water discharge, tunnel valleys increase in number (from 3 to 4) and in dimensions (up to 1.1 cm wide, up to 16 cm long and up to 0.10 cm deep) and keep ensuring the drainage of the distributed water film. Tunnel valleys continue to grow downstream, tracking the advance of the lobate margin and they incise the belt of submarginally-produced ribbed bedforms formed during the previous stage, as indicated by elevation profiles (**Figs. 6c, 10a, d**). The silicone flow velocity decreases along the stream axis, lobe growth slows down ($L_{stream} = 20$ cm; $V_{surf} = 0.02$ mm·s$^{-1}$) and silicone thickness along lobe margin stabilizes between 5 and 15 mm. In response to this slowdown, the lateral velocity gradient decreases across the lateral stream margins, whose width becomes constant ($L_{shear\,margin} = 11$ cm). In these areas, the deformation of the strain ellipse ($\varepsilon_{extending} = + 1.9 \times 10^{-4}$ s$^{-1}$; $\varepsilon_{compressing} = - 1.0 \times 10^{-4}$ s$^{-1}$) and the shear strain rate ($\varepsilon_{shear} = 1 \times 10^{-4}$ s$^{-1}$) stabilize at low values (**Fig. 7c**). New and pre-existing submarginal ribbed bedforms respectively form and grow perpendicular to oblique (70°) to the trunk axis and perpendicular to compressing axes of the strain ellipses (**Fig. 7c, 11c**). Submarginal ribbed bedforms with oblique orientations correspond to bedforms that respectively form and grow during the lobe spreading where the silicone velocity vectors deviate from the trunk axis orientation. Individual submarginal ribbed bedforms increase in dimensions and tend to evolve into a roughly coalescent belt that strike perpendicular to compressing axes of the strain ellipses along the lobe margin (**Figs. 7c, 10c**; 1.4 cm in width, 4.2 cm in length and 0.1 cm in height), although they still display a regular wavelength, from 1.0 to 1.5 cm. New lateral ribbed bedforms form upstream below shear margins with perpendicular orientations to compressing axes of the strain ellipses Elevation profiles reveal they remain smaller in dimensions and display lower relief than their submarginal counterparts (**Figs. 10e**; 2.3 cm in length, 0.7 cm in width and up to 0.07 cm in height). Pre-existing and newly-formed lateral ribbed bedforms are arranged in regular and oblique patterns, with a mean long-axis orientation of 45° to the trunk axis (**Fig. 11c**), and with a wavelength close to 0.7 cm (**Fig. 6c**).

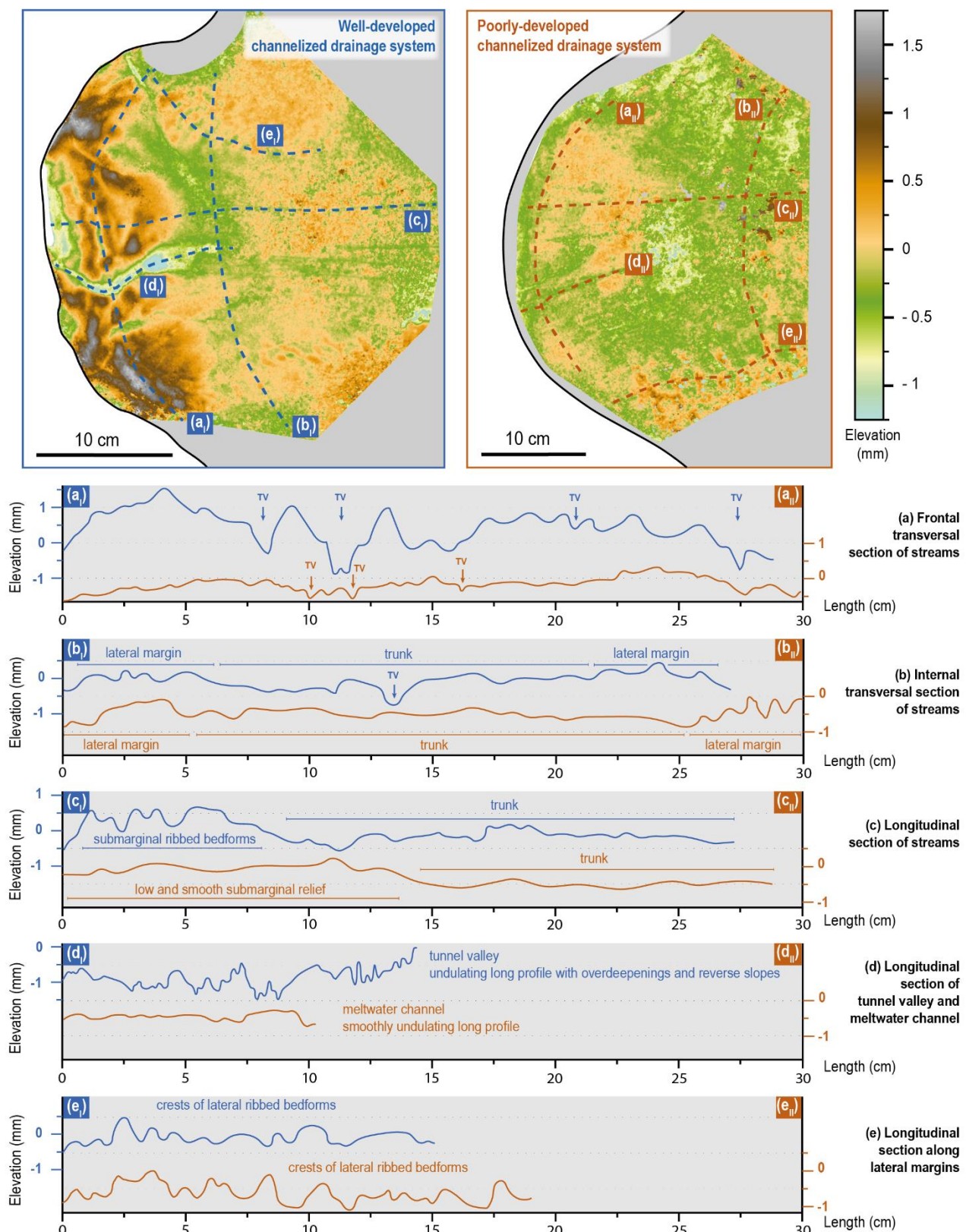

**Figure 10 DEMs of the bed surface beneath the lobes of the silicone streams at the final stage of experiment 10 (blue) and 14 (brown). Ten elevation profiles (a to e) within the lobes are drawn to compare the morphological imprint beneath the two types of streams.**

The grooves still increase in length, but at a slower rate than during previous stages, although remaining subparallel (0-10°) to the silicone flow direction (**Fig. 11c**). The position and length of the grooves correlate with the time-averaged velocity profile measured across the silicone stream (**Fig. 11a**). Hummocky bedforms occur either sparsely between lateral ribbed bedforms or more densely in lobe corners. Their average length is 0.6 cm and their average width is 0.4 cm (**Fig. 6c**).

**Poorly-developed channelized drainage system (experiment 14)**

In response to the imposed increase in water discharge, the sparse, shallow and narrow meltwater channels characterized by smoothly undulating long profiles (**Figs. 10a, d**; n = 9, up to 0.5 cm in width and up to 0.05 cm in depth) are not able to evacuate all the water transmitted to the bed. Thus, the distributed water film, hitherto constrained to the uppermost part of the stream, spreads down to the lobe (**Fig. 8c**). The stream velocity increases and the lobe undergoes a surge (**Fig. 9c**; $V_{surf}$ = 0.15 mm·s⁻¹). The high velocity gradient between the stream and the inter stream area maintains a high rate of deformation of the silicone along the shear margins, with a strain ellipse rotation increasing downstream ($\alpha$ = up to 18°; $L_R$ = 25%) and a maximal value of shear strain rate localized along stream borders ($\varepsilon_{shear}$ = 3 × 10⁻⁴ s⁻¹). As the lobe advances, the stream thins and the silicone column is 20 mm thick along shear margins. The shear margins widen downstream, reaching a maximum width at the silicone cap margin within the lobe corners ($L_{shear\ band}$ = up to 20 cm). Simultaneously with the downstream migration of the water film, the submarginal ribbed bedforms are eroded and evolve into a single shallow and smooth submarginal relief slightly higher than the trunk (**Fig. 10c**). Located beneath the shear margins, the lateral ribbed bedforms keep increasing in number (from 26 to 42) and in dimensions (**Figs. 8c, 10e**; 3.6 cm in length, 1.1 cm in width and up to 0.09 cm in height). New lateral ribbed bedforms form almost perpendicular (70 to 90°) to compressing axes of the strain ellipses, while some pre-existing ones become progressively rotated because of the lobe spreading, resulting in the shift and rotation of the downstream part of the lateral shear margins. Lateral ribbed bedforms are characterized by subnormal to oblique long-axes, deviating by 80 to 40° from the main trunk axis and with a mean orientation of 50° (**Fig. 11d**). They are spaced with a wavelength comprised between 1.5 and 1.9 cm (**Fig. 10e**) and develop on both sides of a flat and grooved trunk (**Fig. 10c**). The corridors of lateral oblique ribbed bedforms thus constitute two bands of topographic highs below the shear margins (**Fig. 10b**).

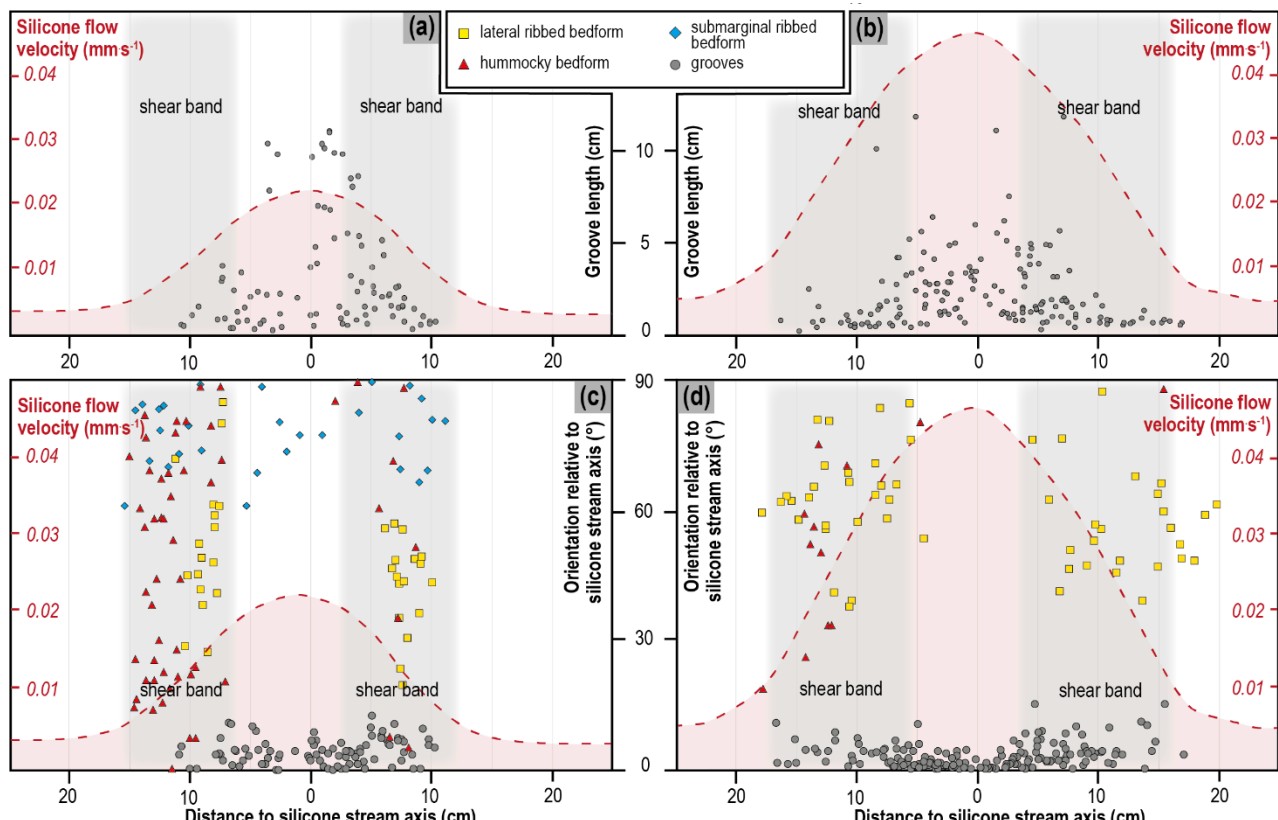

**Figure 11 Morphometric properties of bedforms produced beneath each type of stream at the final stage of Experiments 10 (right panel) and 14 (left panel) plotted according to the distance from the silicone stream central axis. (a) and (b) relationship between groove length and silicone flow velocity (mm·s⁻¹). (c) and (d) variation in orientation relative to silicone stream axis of each population of bedforms (lateral ribs, submarginal ribs, hummocky) compared to the silicone flow velocity and the shear band position (grey columns; defined as $\varepsilon_{shear}$ > 0.5 × 10⁻⁴ s⁻¹).**

The grooves are more elongated near the trunk axis where the highest flow velocities and the greatest cumulative displacement are recorded (**Fig. 11b**). Following the radial spreading of the silicone velocity vectors towards the lobate margin, the groove orientations curve and form a fan-shaped swarm (**Fig. 8c**). Groove orientations mostly remain subparallel to the local silicone flow, with a maximum deviation of 15° from the silicone flow direction (**Fig. 11d**). Sparse hummocky bedforms (width = 0.6 cm; length = 1.0 cm) occasionally occur within the corridors of lateral ribbed bedforms.

### 4.2 Palaeoglaciology: ribbed bedforms beneath ice stream margins

Palaeo-ice stream trunks of the Laurentide Ice Sheet – both with marine (**Figs. 12a-b**) and terrestrial (**Figs. 12c-h**) margins – are characterized by dense swarms of streamlined bedforms (e.g. mega scale glacial lineations, drumlins), evidencing former ice flow directions. Several ice streams were identified, mapped and described by others and compiled by Margold et al. (2015). They were recognized through sharp transitions in streamlined bedform zonation, topographic borders, shear 460 moraines and marginal bedforms. In total, we mapped 303 ribbed bedforms in four distinct areas on three palaeo-ice stream beds.

In the eastern branch of the marine-based Amundsen Gulf Ice Stream (AGIS), the trunk is characterized by streamlined bedforms delimited by two topographic borders. Two fields of ribbed bedforms (n = 62) elongated oblique (mean orientation = 25 to 45°) to the streamlined bedforms occur along the lateral margins. The internal part (i.e. close to the 465 trunk axis) of those two fields is partially overprinted by the streamlined bedforms observed along the trunk. The ribbed bedforms display elongated and arcuate shapes (l/w = 4.2; mean length = 4780 m; mean width = 1110 m), and display a mean wavelength of 1450 m (**Figs. 12a-b**).

In the northern portion of the Hay River Ice Stream (HRIS), broad arcuate to rectangular ribbed bedforms (n = 78) strike oblique to the streamlined bedforms of the trunk. The ridges are clustered in an elongated corridor located between the 470 northern shear margin and the swarm of streamlined bedforms. The shear margin is marked by topographic borders, a sharp transition between a rough inter-ice stream terrain and a smooth trunk covered by lineations, and linear ridges similar to shear moraines. The ribbed bedforms exhibit a mean orientation of 55° to streamlined bedforms and a mean wavelength of 460 m. Those ridges are shorter (1040 m in length and 360 m in width) and slightly less elongated (l/w = 2.9) than those of the AGIS (**Figs. 12c-d**).

In the upstream portion of the Central Alberta Ice Stream (CAIS), oblique and transverse ribbed bedforms (n = 71) are also recognized. Similar to the AGIS and HRIS, the ice stream trunk is characterized by topographic borders, a smoother bed than the surrounding landscape, a meltwater channel and a swarm of lineations. The ribbed bedforms located to the east of the meltwater channel are superimposed by lineations, while the oblique ribbed bedforms along the ice stream margins are apparently not overprinted by other structures. The ribbed bedforms display orientations ranging from 15 to 480 25° to streamlined bedform for the slightly oblique set, while the transverse set displays orientations deviating by 85°. They are 2660 m in length and 690 m in width – thus displaying a mean elongation ratio of 3.8 – and have a mean wavelength of 685 m (**Figs. 12e-f**). The oblique ribbed bedforms are characterized by less arcuate and more elongated shapes than the transverse ones. Downstream and further south in the CAIS, is a widespread belt of transverse ribbed bedforms (n = 92; mean orientation = 95°) overprinted with perpendicular lineations and cross-cut by large meltwater 485 channels (width = 0.6 to 3.4 km). This belt presents a regular pattern of arcuate and coalescent ribbed bedforms (wavelength = 600 to 1 200 m), paralleling a curved belt of fine, linear and almost continuous ridges – similar to recessional moraines – depicted further north. The mean length (4350 m), width (1350 m) and elongation ratio (l/w = 3.0) of the ribbed bedforms show that these ridges are longer, wider but less elongated than the oblique and transverse ridges upstream (**Figs. 12g-h**).

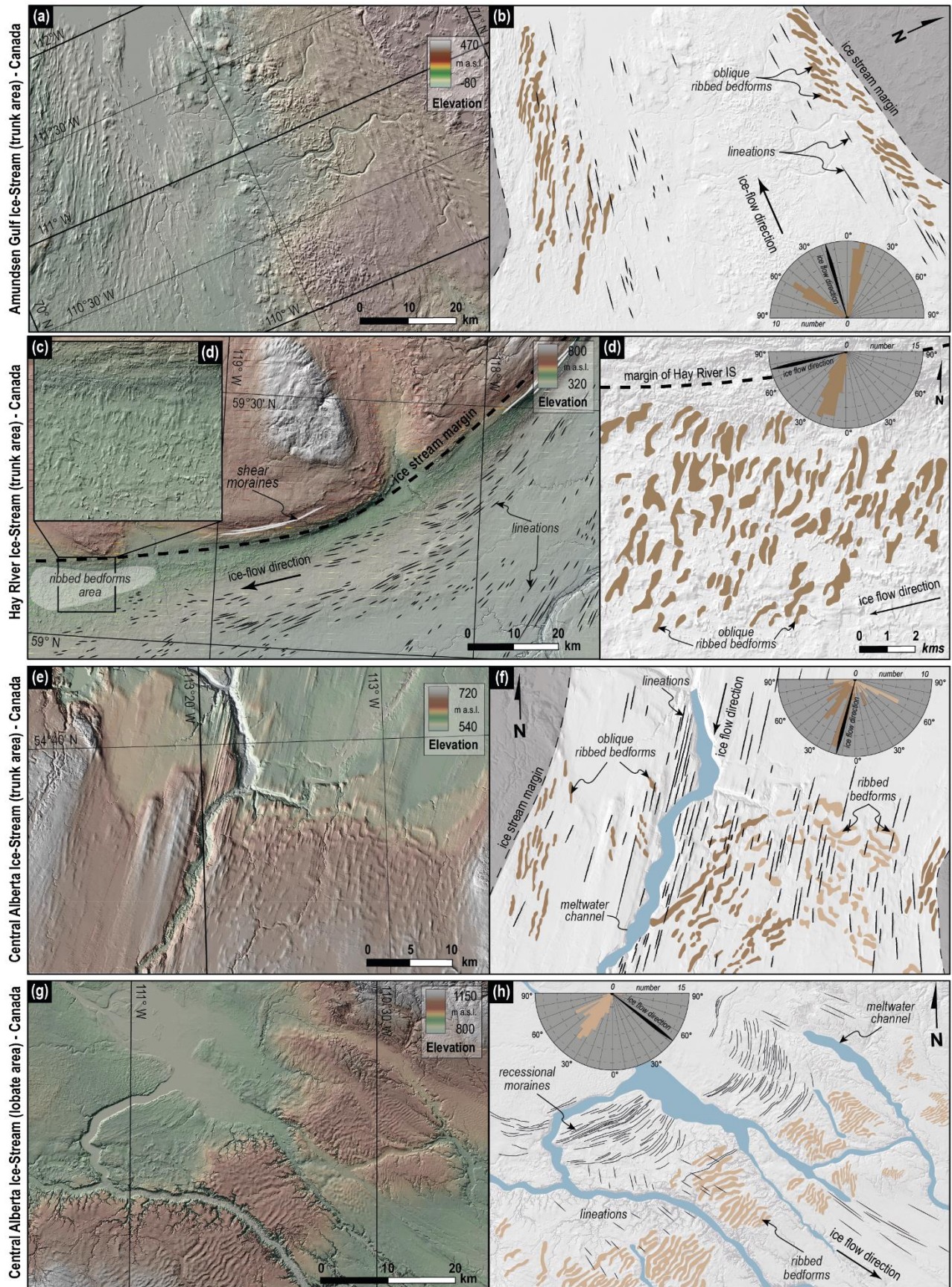

**Figure 12 Manual mapping of ribbed bedforms, lineations and ice stream margins along different palaeo-ice stream tracks in Canada. Left panels show digital elevation models and hillshades using Arctic DEM database (Porter et al., 2018) and Lidar data (Alberta Geological Survey). Right panels show our morphological interpretations and the rose diagrams compiling bedform orientations.**

## 5 Discussion

### 5.1 Morphometric comparisons between experimental and natural ribbed bedforms

Our experiments and observations suggest that ribbed bedforms can develop in lateral corridors and submarginal belts displaying transverse or longitudinal ice velocity gradients, like shear margins and lobes of ice streams, respectively. They occur along the borders of swarms of elongated structures (**Figs. 6, 8, 12**; i.e. lineations in nature and grooves in the experiments) that reveal fast-flowing trunks (Stokes and Clark, 2001; Clark and Stokes, 2003). Beneath the shear margins of the experimental ice streams, the lateral ribbed bedforms form perpendicular to the compressing axis of the strain ellipse. Their mean long-axis orientation strikes at ~45° to the trunk axis and to the boundaries of lateral shear margins (**Figs. 10c-d),** and they tend to gather in corridors elongated parallel to the flow direction. In the experiments, their elongation ratio of 3.3 (mean length = 3 cm; mean width = 0.9 cm) lies in between those of natural intermediate-sized ribbed bedforms considered as the topographic expression of traction ribs, and those of mega-scale transverse ridges, fitting a moderately elongated type of ribbed bedforms. The submarginal ribbed bedforms generated below lobe margins have a mean elongation ratio of 3.0 (mean length = 4.2 cm; mean width = 1.4 cm) and their long-axis is almost perpendicular both to the local silicone flow and the local silicone compressing axis. Experimental lateral and submarginal ribbed bedforms display wavelengths ranging from 7 to 19 mm, and 10 to 15 mm respectively. These wavelengths lie in between 0.4 and 1.5 times the overlying silicone thickness, suggesting that experimental ribbed bedforms tend to form with a spacing slightly lower or equal to the silicone cap thickness, like intermediate-sized ribbed bedforms resembling traction rib patterns (Sergienko and Hindmarsh, 2013). Our observations of ribbed bedforms developing diagonally to the flow and perpendicular to silicone compressing axis in the physical experiments suggests that this might also occur in nature. Examples have been noted in the literature such as oblique ribbed bedforms traction ribs resembling in shape and pattern (Stokes et al., 2016), oblique ribbed moraines with superimposed drumlins (Greenwood and Clark, 2008), transverse asymmetrical drumlins (Shaw, 1983) and drumlins with "en echelon" arrangements (Clark et al., 2018). Along the shear margins of natural palaeo-ice stream beds, we mapped lateral ribbed bedforms whose long-axes display oblique orientations (15° to 55°) to the ice flow direction along palaeo-shear margins (**Figs. 12a-f**), while they tend to display a transverse orientation (~90°) close to lobe margins (**Figs. 12g-h**). The lateral ribbed bedforms have medium elongation ratios (l/w = 2.9 - 4.2), and display an arcuate shape and a regular pattern. Along the shear margins of the AGIS (**Figs. 12a-b**), the obliquity and the apparent overprinting of elongated bedforms was hitherto interpreted as converging and crosscutting generations of streamlined bedforms (Winsborrow et al., 2004; Stokes et al., 2006; De Angelis and Kleman, 2007) and mega-scale transverse ridges (Greenwood and Kleman, 2010), and thus as distinct flow sets arising from a change in flow direction over time. Given that the theory of a continuum in subglacial bedforms emerged in recent works (Stokes et al., 2013; Fowler and Chapwanya et al., 2014; Barchyn et al., 2016; Ely et al., 2016; Fannon et al., 2017), it seems reasonable to propose that bedforms previously interpreted as drumlinized or overprinted ribbed bedforms could form an intermediate bedform between ribbed bedforms and drumlins. Indeed, our experiments demonstrate that streamlined bedforms juxtaposed to oblique ribbed bedforms could form simultaneously in response to the deformation of the strain ellipses, below a single flow set with lateral velocity gradients (**Figs. 6-9**). If it is common for ribbed bedforms to form diagonally to the ice flow and form under the same flow conditions as streamlined bedforms, then this needs careful consideration when using mapped bedforms to plot former ice flow directions such as in flow sets. Based on this hypothesis, we suggest that in the AGIS (**Figs. 12a-b**), these landforms were potentially generated in a single phase under unidirectional flow rather than separate flow sets. An additional argument here is with regard to the elongation ratio obtained from the ridges mapped in this study (l/w = 4.2), and compared to the usual and much higher elongation ratio of mega-scale glacial lineations (l/w = 8.7; Stokes et al., 2013a). Similarly, even though they did not interpret their orientations, Greenwood and Kleman (2010) suggested that these oblique bedforms occurring along the lateral ice-stream

margins could correspond to mega-scale ribbed bedforms (l/w = 5.7).

Furthermore, Sergienko and Hindmarsh (2013) and Sergienko et al. (2014) deciphered rib-like patterns of high basal shear
stress, driven by low hydraulic conductivity, beneath some modern ice streams (e.g. Pine Island Glacier or North-East
Greenland Ice Stream). Because their orientation and formation may be controlled by the ice deformation directions and
not directly by the ice flow directions, rib-like features show clear oblique orientations and symmetrical distribution below
the shear margins with angles deviating from 20 to 70° to the ice flow direction (**Fig. 13**), comparable to the lateral ribbed
bedforms depicted in the AGIS (**Figs. 12a-b**). Regarding the CAIS, the ridges mapped in the upstream (trunk) and
downstream (lobe) areas (**Figs. 12e-h**) were previously interpreted as a type of ribbed bedforms mimicking the pattern of
traction ribs (Stokes et al., 2016) or overridden thrust masses (Evans et al., 2008, 2014; Atkinson et al., 2018). For both
cases (i.e. trunk and lobe positions), we find the exceptional regularity in wavelength, form and scale between ridges to
be suggestive of a common rather than entirely separate origin.

The comparison between experimental and natural ribbed bedforms occurring along ice stream margins is here limited to
the mapping of already identified margins of some palaeo-ice streams. Other natural examples can be found in the south
of the Wollaston Peninsula, in the northeast of Ireland and in the southernmost sector of the Scandinavian Ice Sheet
(Szuman et al., 2021). Despite those observations, the ribbed bedforms presented in this study certainly do not form nor
preserve below all lateral shear margins and frontal lobes of palaeo-ice streams. Further investigation on a large sample
of palaeo-ice stream beds is thus necessary to explore the conditions of preservation and formation of these characteristic
ribbed bedforms along ice stream margins.

Considering their common spatial patterns, orientations, elongations, wavelength/cap thickness ratios and locations, we
suggest that the ribbed bedforms (i) reproduced through experimental modelling and (ii) highlighted in our mapping
within palaeo-ice stream beds are analogous. We also suggest that they represent ribbed bedforms with a medium
elongation ratio like ribbed bedforms resembling traction rib patterns. Our suggestion of commonality between
experimental and natural ribbed bedforms based on morphological characteristics is preliminary and has to be tested in
view of basal interactions and formation processes.

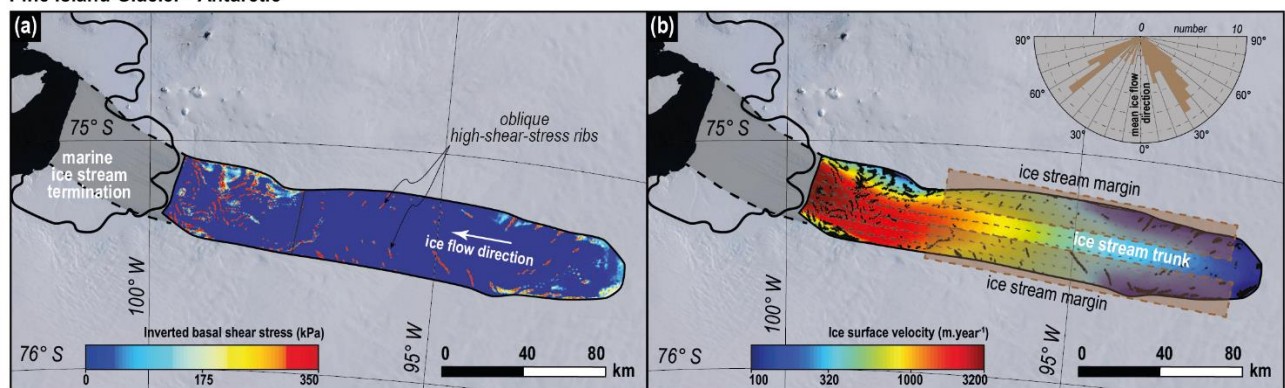

**Figure 13: Maps of inverted (a) basal shear stress (kPa) beneath Pine Island Glacier and (b) ice-surface velocity (modified from
Sergienko and Hindmarsh, 2013). Small and isolated areas of very high basal shear stress (red on (a) and black on (b)) reffered
to as "high-shear-stress ribs" are obliquely set to the main ice flow direction . The orientations of oblique ribs relative to mean
ice-flow direction, located in the brown rectangular areas (along ice stream margins), are compiled in a rose diagram in (b) (©
Landsat / Copernicus - © Google Earth).**

**5.2 Processes of ribbed bedform formation at ice stream margins**

Our experimental results suggest that ribbed bedforms can form below shear and lobe margins of ice streams, areas
experiencing high velocity gradients and transitions in basal water drainage (Echelmeyer et al., 1994; Patterson, 1997;
Raymond et al., 2001). As observed in some of their natural counterparts, we experimentally demonstrated that shear
margins can develop in response to sustained fast ice flow generated by surging events and lubrication of the silicone-bed
interface (Dunse et al., 2015; Schellenberger et al., 2017; Lelandais et al., 2018; Sevestre et al., 2018; Zheng et al., 2019).

In the light of these modelling results, we now propose a process of formation of experimental ribbed bedforms (**Fig. 14**) that takes into account their shape, the overlying cap dynamics and the basal water drainage. We then discuss the formation of experimental ribbed bedforms in the light of existing formation theories.

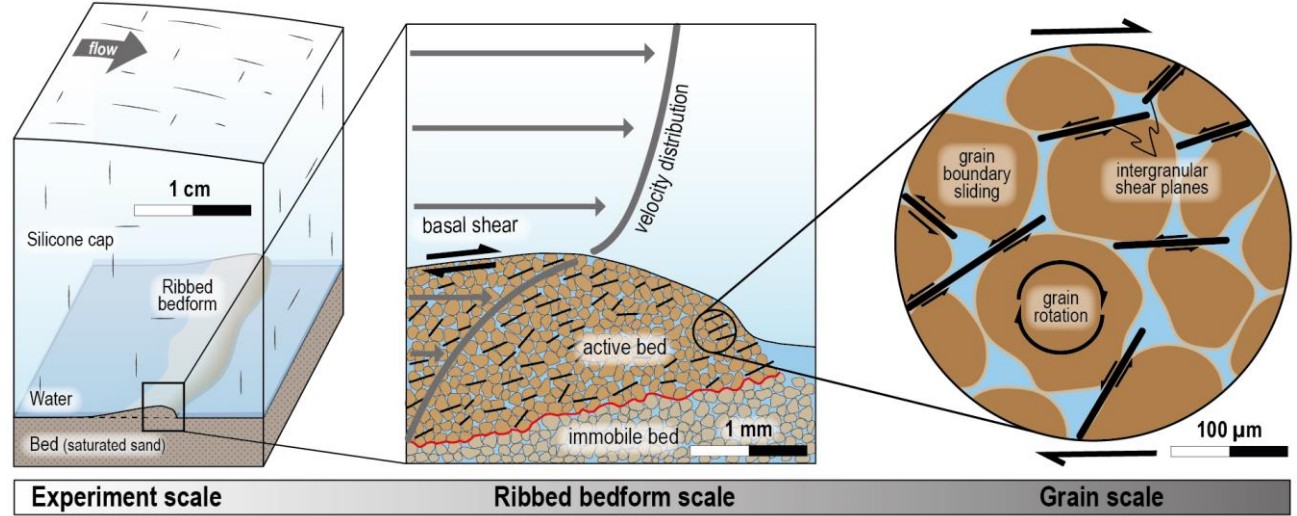

**Figure 14 Model of experimental ribbed bedforms formation. Deformation of a water-saturated and granular bed during silicone-bed coupling. The active bed (i.e. superficial portion of the bed accommodating the basal shear generated by the flow of silicone putty) is able to produce millimetre-thick bedforms through boundary sliding of individual grains along intergranular shear planes and individual grain rotations.**

### 5.2.1 Formation processes of experimental ribbed bedforms

Bougamont et al. (2011) and Tulaczyk et al. (2000) proposed that the formation of a shear margin can be entirely controlled by the subglacial hydrology if the bed properties and the ice thickness remain constant. According to our experiments, we suggest a similar behaviour: the lateral transition between the widespread and pressurized water film (Fig. S3) in the trunk that induces silicone-bed decoupling and the dewatered outer area that induces silicone-bed coupling, generates a stress balance disequilibrium (Fig. 15). In this configuration, the decrease of the basal drag in the trunk area induced by widespread decoupling has to be accommodated by an increase in lateral drag responsible for the formation of shear margins. Margin-parallel simple shear of the silicone occurs along the margins because this region accommodates a high lateral velocity gradient, conditioned by a transition in the basal water drainage (Fig. 15). The combination of silicone-bed coupling and moderate silicone flow velocity (compared to the stream and the outer-stream areas) generates a high basal shear stress along the basal interface and bed deformation below the shear margins. Considering the size of the ribbed bedforms (1mm in height) and the mean grain size of the bed ($d_{med} = 100$ μm), the deformation of the bed must characterized at the granular scale. Boundary sliding of individual grains along intergranular shear planes and individual grain rotations can produce bulk deformation (Hamilton et al., 1968; Oda and Konishi, 1974; Owen, 1987; Bestmann and Prior, 2003). Thus, the basal shear stress may be accommodated by deformation resulting in the genesis of regularly-spaced ribbed and hummocky bedforms (Fig. 14). The water film spreads between the incipient ribbed bedforms to form a linked-cavities system separated by ridges, whose crests remain coupled to the overlying silicone and keep accommodating basal shear stress. Because of the continuous deformation and flow by simple shear, the instantaneous strain ellipse in the silicone deforms and strain axes form at 45° to the shear margin boundaries while the amount of shear strain increases (Fig. 15). Given that lateral ribbed bedforms form beneath the shear margins where the silicone cap is coupled to the bed, their long-axes strikes at 45° to the flow direction.

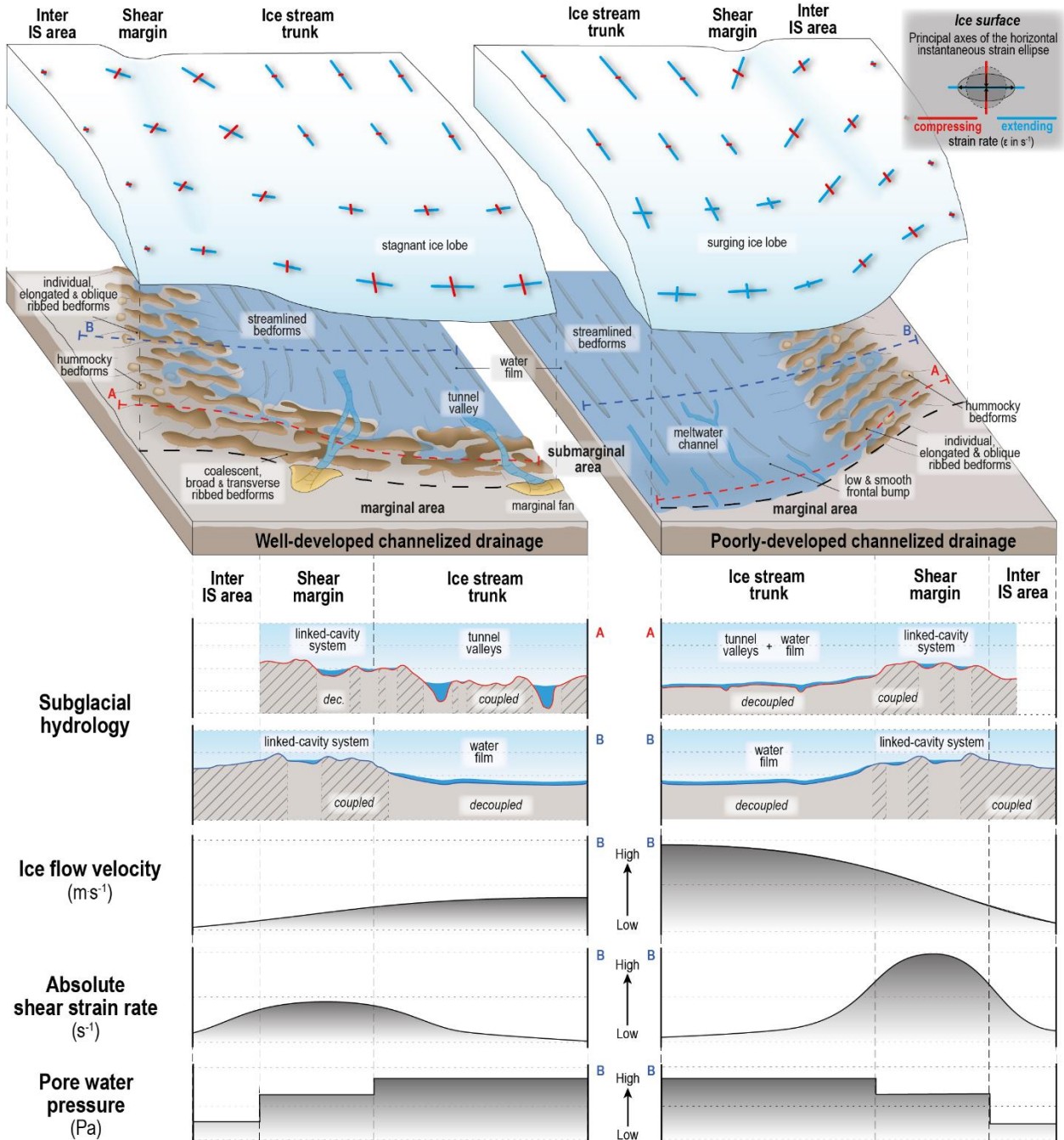

**Figure 15 Conceptual model summarising our observations and interpretations and the proposed role of hydrology, ice velocity and superficial ice strain in generating the observed landforms and their spatial distribution. This is for a terrestrial ice-stream context either characterized by well-developed or poorly-developed channelized drainage below the lobe frontal margin.**

Beneath lobe margins, the water film channelizes into a system of a well-developed tunnel valleys resulting in basal coupling and high basal shear stress in interfluve areas (**Fig. 15**). This reduction in basal lubrication leads to a slowdown in margin advance, a stabilization of the ice lobe and a downstream increase of longitudinal compressive stress. Similar to lateral ribbed bedforms, the high basal shear stress below lobe margins might be periodically accommodated by bed deformation. Once initiated, the submarginal ribbed bedforms continuously grow and elongate perpendicular to the compressing axes of the strain ellipses (i.e. parallel to the lobe margin) and a linked-cavities system develops in between the set of coalescent submarginal ridges (**Fig. 15**). Belts of coalescent ribbed bedforms mirroring the shape of the lobe margin can also form subglacially in the distal part of the lobes. Similar coalescent ridges close to lobe termination have been hitherto interpreted as landforms resulting from the bulldozing of marginal sediments or the subglacial remoulding of former marginal landforms as a result of lobe re-advances (e.g. glaciotectonic thrust masses, overridden moraines and push moraines for examples; Totten, 1969; Boulton, 1986; Benn and Evans, 2010; Evans et al., 2014). We here propose

an in-situ and self-organized formation process for ribbed bedforms resulting from the shearing of the sedimentary bed (i.e. till layer or weak bedrock) beneath lobes.

**5.2.2 Comparison between experimental bedform formation and current theories**

The consistent wavelength of natural and experimental ribbed bedforms described in this paper and their similarity in form, orientation, location and scale supports a common model of formation.

Several theoretical models have been established for the development of subglacial bedforms. The large-scale subglacial meltwater flood hypothesis (Shaw, 2002) does not correspond to the process that occurs in our experiment since silicone flow, rather than water flow, is responsible for their formation. Indeed, experimental ribbed bedforms only initiate in zones of silicone-bed coupling along the lateral margins of the flood path. Except for Shaw's theory, all theories of formation of subglacial ribbed bedforms are associated to a till or a sedimentary substratum being coupled to an overlying ice sheet that flows and either deforms or remoulds its bed. Even though the recycling of marginal landforms (i.e. marginal deltas and pushed sediments) during the silicone advance might be involved in the growth of submarginal ribbed bedforms in the experiment, the pre-existence of ridges or mounds is not a pre-requisite for their formation (Boulton, 1987; Lundqvist, 1989; Möller, 2006). Hättestrand and Kleman (1999) related the development of ribbed bedforms to transitions in the basal thermal regime and to the fracturing of a frozen bed. This cannot occur in our experiment, because thermal and melting/freezing processes are not simulated.

Others hypothesis have sought specific mechanisms to explain the formation of ribbed bedforms that are neither related to meltwater flood, nor pre-existing ridges or frozen bed. Some view ribbed bedforms as natural instabilities resulting from the coupled flow of ice, meltwater and deformable till, and conditioned by spatial variations in effective pressure, basal thermal regime and bed strength (Hindmarsh, 1998a,b; Fowler, 2000; Schoof, 2007; Dunlop et al., 2008; Chapwanya et al., 2011; Fowler and Chapwanya, 2014; Fannon et al., 2017). Others have suggested that bed deformation can form transverse bedforms through shearing and stacking of sediments, resulting from the flow of ice over the bed interface (Shaw, 1979; Aylsworth and Shilts, 1989; Bouchard, 1989; Lindén et al., 2008; Stokes et al., 2008). Both processes, regardless of the bed rheology, involve a deformable, flat and temperate bed whose deformation results from the basal shear stress induced by the overlying ice. The high basal shear stress and the associated bed deformation are supposed to enable the formation of periodic subglacial ribbed bedforms transverse and oblique to the ice flow direction, either below modern ice streams (Sergienko and Hindmarsh, 2013; Sergienko et al., 2014) or along palaeo-ice stream beds (Stokes et al., 2016). The bed in our experimental model is flat and temperate, and deforms under the action of the basal shear stress induced by the flow of the silicone putty. The model demonstrates that such periodic ribbed bedforms could form below shear and lobe margins of streams. Consequently, the process responsible for the formation of experimental ribbed bedforms is very similar to the current theories that view natural ribbed bedforms as a product of till deformation under the effect of high shear stress along the basal interface (**Fig. 14**). However the technical limits of the experimental set-up do not allow for these theories to be physically tested, since we cannot estimate the water and effective pressure at the scale of a single ribbed bedform.

The contemporary development of ribbed bedforms and linked-cavities systems we observe in the experiments is consistent with the sedimentology work carried by Linden et al. (2008) relating the development of ribbed bedforms to the deposition of sediments into the lee-side cavity of moraine ridges. Shear-heating within natural shear margins tends to produce high melting rates and subglacial water flow that could favour the development of water drainage systems (e.g. linked cavities, meltwater channels) in-between ribbed bedforms (Perol et al., 2015; Perol and Rice, 2015). Hummocky bedforms, which have hitherto been considered mostly as marginal landforms associated to ice stagnation and retreat (e.g. Johnson and Clayton, 2003), seem to exhibit a spatial association and a genetic proximity with the ribbed bedforms below the experimental shear margins. Some studies suggest that a morphological continuum between quasi-

circular bedforms (i.e. hummocky shape) and ribbed bedforms exists below ice streams and ice sheets (Stokes et al., 2013a; Ely et al., 2016). It is thus appropriate to hypothesize a common origin for some hummocky bedforms coexisting with fields of ribbed bedforms (**Fig. 15**). They could be alternately interpreted as proto-ribbed bedforms in a continuum of subglacial bedforms inferred from the evolution of the basal shear stress intensity.

The experimental results provide an array of bedforms and contexts that can be used as analogs for ice stream landsystems.

Our work faithfully reproduces the distal/lobate part of ice streams since channelized features, marginal deltas and pushed sediments are observed. However, some key bedforms observed beneath natural ice streams are not reproduced in the experiment. The analogue model does not reproduce streamlined bedforms that resemble drumlins or mega-scale glacial lineations (MSGLs). If the process that forms experimental ribbed bedforms is identical to those of current till deformation theories, which form both ribbed and streamlined bedforms, our experimental model should allow the formation of

drumlins and MSGLs. Streamlined bedforms as opposed to ribbed bedforms, are mostly explained as formed beneath corridors of high ice flow velocity (Clark, 1993; Stokes and Clark, 1999; Stokes et al., 2013). In our experiments, we inferred that ribbed bedforms form where the silicone-bed interface is coupled and where silicone flows 10 times faster than the surrounding stagnant silicone cap. By comparison, the trunk of silicone streams – where silicone flows the fastest and ribbed bedforms do not form – exhibit velocities up to 30 times faster than the surrounding silicone cap. As the only

way to trigger an experimental silicone stream is to initiate decoupling above a pressurized water film, the basal interface of silicone stream is thus entirely decoupled, therefore providing an explanation for the lack of drumlins and MSGL in our experiments. Where the silicone-bed interface is coupled, which is one of the key conditions to form streamlined bedforms (Boulton, 1987; Hindmarsh et al., 1998; O Cofaigh et al., 2005; Fannon et al., 2017), the maximum silicone flow velocity we experimentally reproduce is equal to the silicone flow velocity where ribbed bedforms form; thus it is

probably insufficient to initiate streamlined bedforms. Indeed, the presumed velocities that enable drumlins and MSGLs formation in nature are 10 or 100 times faster than where ribbed bedforms form, respectively (Stokes et al., 2013). Furthermore, the low viscosity of the silicone putty and its potential high rates of creep closure prevent the formation of water-free cavities, crevasses, Nye-channels and the downstream propagation of basal ice roughness that have sometimes been proposed as prerequisites for the formation of drumlins, MSGLs and shear moraines (Bluemle et al., 1993; Punkari,

1997; Tulaczyk et al., 2001; Clark et al., 2003; Smith et al., 2007).

### 5.3 Ribbed bedforms in ice stream landsystems

### 5.3.1 Implications for reconstructing the spatial organisation of ice streams

Ribbed bedforms are ubiquitous and conspicuous features covering extensive areas of former ice sheet beds (Aylsworth and Shilts, 1989; Hättestrand and Kleman, 1999; Souček et al., 2015; Stokes, 2018). Their occurence along palaeo-ice

stream beds and their relation with the dynamics of fast-flowing ice corridors have however been little discussed since their formation has mostly been attributed to slow ice flow. Ribbed bedforms have been described so far mostly in isolated patches in trunks (Stokes et al., 2008) or in large fields in the onset zone of ice streams (Dyke et al., 1992), and more rarely in corridors (e.g. ribbons track; Dunlop and Clark, 2006; Trommelen et al., 2014) or belts (Greenwood and Kleman, 2010; Stokes et al., 2016) like the ones we mapped on DEMs and remotely-sensed images or identified in the experiments.

The coexistence of ribbed bedform patches (i.e. sticky spots) within swarms of streamlined bedforms and alternating corridors of ribbed and streamlined bedforms below ice streams have previously been reported. Ribbed bedforms coexisting with streamlined bedforms have been associated to velocity gradients resulting from the spatial and temporal variations in the distribution of slippery and sticky portions of the bed (Shaw, 1979; Bouchard, 1989; Lindén et al., 2008; Stokes et al., 2008, 2016; Trommelen et al., 2014). By combining experimental and natural observations, we demonstrate

that velocity gradients at ice stream and lobe scale could be identified through specific patterns of ribbed bedform

distribution. We therefore hypothesize they could constitute additional morphological markers contributing to the identification of shear and lobe margins (**Fig. 14**).

These results could have new implications for the identification of palaeo-ice streams. An important point, if our conclusions hold, is that parallel ridge sequences mirroring the shape of lobate margins might be subglacially-produced ribbed bedforms rather than proglacial thrust tectonic ridges, placing the ice margin in a different location in an ensuing reconstruction. Topographic borders and shear moraines are usually used to define the position of palaeo-shear margins (Dyke and Morris, 1988; Dyke et al., 1992; Stokes and Clark, 2002b). Corridors of oblique and regular-spaced ribbed bedforms could be used as an additional proxy for the reconstruction of shear margin positions, notably when shear moraines are not observed or when ice streams are not controlled by the topography. In addition, multiple activations, migrations or retreats of ice streams imply the superimposition of several generations of bedforms that makes ice stream beds a complex palimpsest to decipher (e.g. Clark, 1999). Indeed, the belts of marginal bedforms mimicking the morphology of palaeo-ice lobes and swarms of streamlined bedforms characterizing the position of palaeo-trunks are commonly modified through bed erosion, deformation and overprinting, thus altering partially to fully the initial landsystems. In the light of our results, the interpretation of some fields of ribbed bedform forming successive corridors (parallel to ice flow) or belts (perpendicular to ice flow) deserves reconsideration. Depending on the orientation of the ribbed bedform long axes relative to the direction of streamlined bedforms, as well as the elongation ratio and the spacing between individual bedforms, the corridors and belts of ribbed bedforms could indicate lateral and/or frontal migrations of palaeo-ice stream positions through time. Thus, this study provides additional mapping criteria to constrain the temporal migration of previously-identified palaeo-ice streams (e.g. CAIS; **Figs. 12e, f**). It is important to note that depending on their size, corridors of ribbed bedforms could also illustrate transverse velocity gradients within ice streams, potentially across ice bands flowing at distinct velocities due to variations in basal stick-slip conditions and subglacial roughness.

This hypothesis needs to be tested through the identification of corridors and belts of ribbed bedforms in order to establish if they could correlate with hitherto non-recognized palaeo-ice streams margins. If this hypothesis is validated, ribbed bedforms with specific shapes and orientations could potentially help identify new palaeo-ice streams through the identification of their margins, particularly when features such as MSGL are poorly-preserved or absent.

**5.3.2 Implications for ice stream dynamics and subglacial hydrology**

Our experimental model does not reproduce thermo-mechanical feedbacks within the ice, which have been suggested as the first mechanisms responsible for the self-organization of ice streams (Payne and Dongelmans, 1997; Hindmarsh, 2009). Despite this lack of thermally-based mechanisms, streams initiate in the model as a result of interactions with the basal hydrology as suggested by Winsborrow et al. (2010) and Kyrke-Smith et al. (2014). The combination of analogue modelling – which demonstrates that ribbed bedforms can form through the interaction of silicone flow, water flow and bed deformation – and ice stream bed mapping, has allowed us to establish a link between bedform development, ice stream dynamics and the evolution of spatially and temporally efficient drainage systems (**Fig. 14**). The efficiency of the subglacial drainage system is estimated by the capacity of channelized features (e.g. tunnel valleys and meltwater channels) to accommodate the meltwater discharge (Moon et al., 2014). Therefore, when considering the relationships between the distribution of subglacial water, the development of channelized features and the development of ribbed bedforms below experimental lobes, we suggest that the presence or absence of submarginal ribbed bedforms along lobate margins might reveal the type and efficiency of meltwater drainage below marginal lobes of terrestrially-terminating ice streams (Raymond, 1987; Patterson, 1997; Kim et al., 2016). For instance, a well-developed channelization facilitates subglacial drainage by concentrating meltwater flow into tunnel valleys and induces an overall increase of basal drag beneath the lobe due to water pressure reduction, widespread ice-bed coupling and elevated basal shear stress. The

increase in basal drag, highlighted by the development of submarginal ribbed bedforms, is responsible for ice stream slowdown and stabilization of the marginal lobe. This configuration should be associated with the formation of well-developed end moraines (**Fig. 14**), because the margin experiences a standstill over time. Conversely, a poorly-developed channelization characterized by shallow and narrow meltwater channels reveals the incapacity of the subglacial drainage system to evacuate all the meltwater efficiently. This configuration is more likely characterized by higher water pressure, widespread ice-bed decoupling and distributed drainage that promote basal sliding and inhibits the formation of ribbed bedforms in sub-marginal environments. The inefficient drainage might favour the storage of meltwater up-ice during periods of increased melting that is episodically delivered to the margin through outburst floods. We suggest that outburst flood events and basal sliding could alter the dynamics of ice streams and trigger surging of ice lobes. The absence of submarginal ribbed bedforms and a poorly-developed drainage system should therefore be associated with typical surge-diagnostic features dominated by compressional structures (**Fig. 14**). Glaciotectonic thrust masses and crevasse-squeezed ridges have notably been observed in southeast Alberta and linked to the dynamics of surging lobes during melting and retreat of the Laurentide Ice Sheet (Evans et al., 2020).

## 6 Conclusion

Despite the ubiquitous and extensive nature of subglacially-produced ribbed bedforms beneath former ice sheets (variously called Rogen moraine, ribbed moraine, ribbed bedforms, mega-scale transverse ridges), their significance in ice stream landsystems and their formation processes remain poorly understood. Providing new constraints on the formation, evolution and distribution of ribbed bedforms is therefore critical to characterize ice-bed interactions beneath key zones of ice streams and reconstruct past glacial dynamics. Based on experimental modelling and geomorphological mapping of natural ice stream beds, we suggest that ribbed bedforms are produced subglacially beneath ice streams margins where the soft bed is coupled to the ice and subject to high basal shear stresses that results from abrupt lateral and longitudinal variations in subglacial drainage characteristics and ice flow velocity. We suggest that these ribbed bedforms develop both (i) in narrow corridors parallel to lineations and (ii) in broad belts parallel to and upstream of marginal landforms. The former (i.e. lateral ribbed bedforms) can be used to highlight the position of ice stream shear margins. These ribbed bedforms are regularly-spaced, slightly arcuate and moderately elongated (l/w = 3 to 4). Due to the specific stress and strain configuration, lateral ribbed bedforms developing beneath shear margins of ice streams should present higher elongation ratios than the classic ribbed/Rogen moraines that occur outside ice stream contexts. In our experiments the long axis of ribbed bedforms develops perpendicular to the compressing axis of the strain ellipse and at 45° to the shear margin boundary. This oblique relationship has implications for those reconstructing palaeo ice flow directions (flow sets) from bedforms, where misinterpretation could be made using the usual assumption of ribs forming orthogonal to the flow. Rather than reflecting the flow direction, we demonstrate that the ribbed bedforms are compressional structures whose orientation depends on the orientation of strain axes. The latter (i.e. submarginal ribbed bedforms) form larger belts of coalescent and broad ribbed bedforms characterized by arcuate crests arranged orthogonal to the compressing axis. They are interpreted as the morphological imprints of ice lobe positions, but unlike the usual belt of marginal landforms hitherto described in the literature and interpreted as proglacial glaciotectonic structures, this type of ribbed bedforms initiates and develops subglacially from the basal shearing of a flat subglacial bed. Their preferential development in interfluve areas between meltwater channels implies that their formation is enhanced in zones of channelized and efficient drainage where extensive areas of ice-bed coupling exists due to the decrease of subglacial water pressure. We therefore suggest that those two types of ribbed bedforms (i.e. lateral and submarginal) result from the same deformation process of the underlying soft bed.

These results provide new criteria for palaeo-glaciological reconstructions of ice streams to help identify lateral shear and

frontal lobes and provide insights on ice-bed interactions, ice dynamics and subglacial hydrology. Whether the process

of ribbed bedforms discussed here in the context of ice streams and lobes is relevant and applicable more widely to the

full population of ribbed / Rogen moraines found extensively across many ice sheet beds, is still an open question.

*Author contributions.* ER conceived this research project with contributions by OB, SP and CC, and gathered funding.

RM developed the experimental lab and contributed to the computation of strains in the experiments. TL, DP, ER and

790 RM designed the experimental device. JV and ER conducted the experiments. JV post-treated the experiments, made the

palaeo-glaciological mapping and wrote the first draft of the paper. All authors contributed to the interpretations of the

results and to the proofreading of the paper.

*Competing interests.* The authors declare that they have no conflict of interest.

*Acknowledgements.* This study is part of the ICE COLLAPSE project (Dynamics of ice sheet collapse in deglaciation

periods) funded by the French "Agence Nationale de la Recherche" through grant ANR-18-CE01-0009. This project has

benefited from the PALGLAC team of researchers and received funding from the European Research Council (ERC)

under the European Union's Horizon 2020 research and innovation programme (Grant agreement No. 787263).

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
