# Peer review of "Formation of ribbed bedforms below shear margins and lobes of palaeo-ice streams"

_The Cryosphere, 2020_

## Referee Comment (RC1) · Richard C.A. Hindmarsh (Referee) · 20 Dec 2020

**Review of "Ribbed bedforms in palaeo-ice streams reveal shear margin positions, lobe shutdown and the interaction of meltwater drainage and ice velocity patterns" by Jean Vérité and ten others for *The Cryosphere**

Richard C.A. Hindmarsh

British Antarctic Survey and University of Durham

December 20, 2020

**1 Overview**

This paper by Jean Vérité and colleagues (V++) from *Le Mans Université*, *Université de Nantes*, University of Sheffield and Alberta Geological Survey addresses a centuries-old topic in glacial geomorphology, namely the mechanisms operating and conditions needed to produce certain glacial geomorphological landforms (ribs and drumlins); they focus on ribs transverse to the ice flow direction. V++ adopt an approach of performing laboratory experiments; the authors give the impression of being good and experienced at these and are also very enthusiastic. They use an integrated approach, simulating the ice sheet with a pancake of flowing silica gel, which overlies sand, and pump water in underneath in order to lubricate the base. The tray on which they do this is 2m×2m, and the ice sheet, at least at the start of the experiment, has a roughly circular profile. They report on fifteen of these experiments, with conditions varied. Presumably for practical purposes, they do not include thermal effects in the experiment, in particular lab representations of the transition between ice and water.

During the experiments, water is pumped in underneath the silica gel, at a central location: the higher water pressure causes the silica and sand to disassociate, and for silica streams to form, which V++ compare with ice streams. V++ are aware of the two styles of sub-glacial drainage that have received greatest attention, 'R-channels' (Röthlisberger, 1972) and 'linked cavity systems' (Kamb et al., 1985) and identify both as forming in distinct experiments. R-channels form dendritic systems, while 'linked cavities' are networks.

The title of the paper "...reveal shear margin positions, lobe shutdown and the interaction of meltwater drainage and ice velocity patterns" indicates that they wish to focus on some previously underconsidered aspects of geomorphology. The reasons for these choices become clearer as one reads the paper; the lab experiments produce rib forms in what glacial geomorphologists might regard as unusual locations, and V++ wish to emphasise the spatial realism of their results. This of course a very reasonable approach, but the cautious reader might well start to wonder whether the processes involved in the creation of structures in the lab are the same as those which produce structures in the field. V++ should pay rather more attention to this, considering the detailed physics of both situations.

Glacial geomorphology has undergone quite a substantial change in the past quarter-century; previously it was carried out by geologists and geographers, who did not consider in detail the physics and mathematics of the processes. In the late 1990s theoretical work was done on the coupling between ice and deforming sediment beneath, which turned out to make quantitatively accurate predictions of rib-spacing (though successful models of drumlin formation had to wait a decade or more). This initial work was done primarily and independently by Hindmarsh (1998a,1998b), Fowler (2000) and Schoof (2007) (these papers abbreviated *HFS*), with substantial later contributions from the 'Fowler-school' - Chapwanya, Katz, Kyrke-Smith and Fannon.

Despite its successes, this work has not been widely accepted, owing to its use of a viscous rheology to describe the deformation of till; laboratory experiments carried out in the 1930s by Terzaghi and in the 1990s by Kamb indicated that reproducible results in the laboratory could only be obtained with a plastic rheology. However, the plastic rheology has yet to produce a theory that predicts landform formation such as ribs or drumlins. As Fowler (2010, p.970) puts it "*This suggests that the simplest conceptual model for till deformation is already much more complicated than either a viscous or perfectly plastic material, and that, where till is concerned, there is still a great deal of theoretical work to be done concerning the sliding law*". There is a debate in glaciology about whether till is plastic or viscous, and the answer seems to be that it is more complicated than this opposing pair approach admits.

V++ rather steer clear of the viscous/plastic debate; they cite the *ur*-viscous paper (Boulton and Hindmarsh, 1987) but cite nothing by Terzaghi nor by Kamb. They have plentiful descriptions of glacial geomorphological features (probably over-plenty); the point of this seems to be that their lab work produced various features, which resemble glacial geomorphological features, and they wish to bring this to the attention of the reader. Certainly, their work does produce flow-transverse ribs, but does not produce features aligned with flow that resemble either drumlins or mega-scale glacial lineations (MSGL, Clark 1993). This raises the question of why do their experiments not produce the whole gamut of glacial geomorphology? and brings us back to the question above about the physical realism of their simulations: are their laboratory ribs formed by the

same set of processes that form sub-glacial ribs?

In V++'s favour is that in their experiments ribs are formed. However ribs are formed in nature by a vast variety of processes, for example sand ripples in streams and transverse cloud patterns; in both cases turbulence is involved, which is not held to be a major component in the formation of sub-glacial ribs. This widespread occurrence of ribbing leads the reader to wonder whether the ribs in V++ are produced by the same basic set of mechanisms as those operating beneath ice sheets and glaciers.

V++ presents novel exciting experimental work which will inspire a very large portion of the glaciological community, and should certainly be published, but I don't believe it to be publishable in its present form. The glacial geomorphology descriptions should be reduced substantially; I don't believe that, compared with their length, they contribute anything amazingly new. Rather, V++ should make the points that ribs, drumlins and MSGL exist, and V++ can *simulate* rib-formation; I emphasis 'simulate' because the authors do not make the case that their experimental formation processes represent precisely the same set of morphological processes as are operating beneath glaciers and ice streams. To this end, more attention to the basic physics is required; how did V++ estimate ice pressure, water pressure and effective pressure? From their plentiful quotations of theoretical work by the viscous-till school it seems that they do not oppose this idea on fundamental grounds, so V++ should include more detailed analyses of how their observations of the component pressures relate to *HFS*.

My ideal paper form is §1: Introduction (much as it is now); §2 - review of glacial geomorphological features produced under the ice, with an emphasis on rib descriptions (perhaps contrasting Rogen moraine and traction ribs) and on the viscous (and other) theories of rib formation; §3 - a description of the experimental set-up; §4 - a discussion of how the experimental set-up permits and/or disallows experimental observations that confirm existent theories; §5 - the results; §6 - a discussion of how rib-formation locations are related to ice-stream plan-geometry; §7 - Summary and conclusions. I have little doubt that the authors of V++ will disagree with some of these aspects, but they should recall that the main contribution of the paper is the experiments that they have carried out with panache, and emphasise that their results are largely *consistent* with (but not exactly identical to) nearly two centuries of geomorphological observations. Another way of putting this is that **none of their observations are never found in the field**, but that their silica-sand-water system might have different statistical characteristics from the ice-sediment-water system.

**2   Major Points**

The following points are thoughts that I had during the review; quite a few of them don't include specific suggestions to the authors. I think that all the sentences ending with a "?" need to be given consideration in the revision.

1. V++ §2 reads like a review paper on the glacial geomorphological features shaped by local glacial erosion and deposition (ribs, drumlins and MSGL). It is effectively a catalogue of such forms with little detectable relevance to the main purpose of the paper, which is to present the experimental results. Since the experiments did not produce features resembling drumlins or MSGL, the revised §2 should focus on ribs. A newish feature of the analysis in V++ is the association of rib-field locations with particular locations in ice streams and ice sheets. This possibly leads V++ to overfocus on these - for example rib co-location with stream lateral margins and downstream ends of surge lobes - and ignore the widely agreed observation that rib fields are found upstream of drumlin fields, under slower-flowing portions of the ice sheet. Do V++'s results explain these?

2. Significant work on relating theory and observations of enormous sets of subglacial ribs was done by Dunlop and Clark (2006) and Dunlop at al. (2008) (in fact Chris Clark is one of authors of V++). One of the conclusions of Dunlop et al. (2008) was that their results, obtained from analysing $2 \times 10^4$ ribs, did not falsify the viscous theory of rib formation. Fowler (2000) and Schoof (2007) showed that the dependence of deformation rate on effective pressure (difference between the load exerted by ice and water pressure) was central to understanding how the instability arose; a particular point is that the dependence of flux on effective pressure affects the distribution of negative flux gradients (where the till is thickening). I would like to see more consideration of how V++'s mechanisms of rib formation might be explained in terms of the Fowler-Schoof conditions.

3. There is considerable focus in recent literature on the horizontal dimensions of the landforms compared with the thickness of the ice. If the horizontal dimension is less than or comparable with the ice thickness, the full Stokes equations need to be solved in order to calculate the normal stress exerted by the ice on the sediment accurately; this is needed to calculate the effective pressure. I encourage V++ in their resubmission to provide data on the rib spacings and how this compares with the silica gel thickness at time of rib formation. They could also comment on how their observations coordinate with the Fowler-Schoof conditions for (geo-)morphological instabilities to exist.

4. I recognise that in the past decade two 'species' of ribs have come to be recognised, the long-established ribbed/Rogen moraines with spacings of 300 - 1000 m ($\leq$ ice thickness), and the newer larger 'traction ribs' (Sergienko and Hindmarsh, 2013; Stokes et al., 2016) with spacings of a few kilometres ($\geq$ ice thickness); these traction ribs can be found underneath ice streams in non-traditional rib locations. It is almost certain that modelling traction ribs requires solution of the Stokes equations, despite their large horizontal dimensions, owing to the slippery beds (low 'traction number') beneath streams (Hindmarsh, 2004; Schoof

and Hindmarsh, 2010).

5. A substantial proportion of theories of rib-formation (e.g. Hättestrand and Kleman, 1999) focus on the freezing-melting boundary map-location as a control on rib formation. V++'s experimental set-up does not permit investigation of this aspect, but this matter does require some comment, in particular on the issue of whether, sub-glacially, there is one and only one means of forming ribs. My personal belief is that there may be several.

6. There doesn't seem to be a great emphasis on the fault and fold structure within and surrounding the ribs - a good deal of work on this matter (in glacial landforms) has been published e.g. *Hart et al.*, (1990, Figs. 3&4), *Eyles*, (1993, Figs 3.5&3.6). I appreciate that the sizes of the features will be rather different (V++ order millimetres, geologists order tens of centimetres) and that it is probably not possible to look at the lab-structures now, but some insight might have been gained during the experiments. Were 'faults' observed, and what does this tell us about the styles of deformation?

7. Some thought needs to be put into explaining why the experiments do not produce flow-aligned features (drumlins/MSGL) in the context of work by the Fowler-school modelling of drumlin formation. I appreciate that a definitive answer may not be available yet, but this would be of considerable benefit to those wishing to extend and elaborate the work of V++.

8. Since glacier linked-cavities rely on cavity formation and hydraulic links between the cavities, V++'s association of laboratory-observed networks with 'linked cavities' is quite reasonable, but R-channel theory relies heavily on the heat production by flowing water melting the tunnel that is being closed by the weight of the ice; thermal effects are not included in V++'s experiments. A related point is that R-channel theory and linked-cavity theory have opposite relations between the system transmissibility (product of permeability and vertical area) and effective pressure; R-channels have transmissibility *increasing* with effective pressure, while linked cavities have it *decreasing*. It is not clear whether the dynamics of the sub-silica drainage system are the same as the sub-glacial; for example, might not the lab drainage system development be due to a Hele-Shaw instability? It might be that V++ wish to point out the similarities between the mechanisms of their lab-formed streams and streams in the field, but the real question is whether sufficient observations have been made in either case; I'm pretty sure that not enough is known about stream-formation in nature.

9. In particular, questions were raised in my mind about the mechanisms in the V++ experiments by which the streams were formed. Agreed that there are theories in

which streams are formed through the interaction of ice flow and water flow (via effective pressure) - Google on "Kyrke-Smith", "Katz" for a lead into this - but the first quantitatively-identified mechanisms were through thermo-viscous feedbacks (e.g. MacAyeal, Payne, Hindmarsh in the '90s). As mentioned above, V++ do not include thermally-based mechanisms in their experiments, which leads naturally to wondering about their lab-produced ribs adjacent to stream boundaries - is this saying (as they seem to be suggesting) that one condition for rib formation is a large lateral velocity gradient - a glaciological insight of potentially great importance - or are there some special thermal characteristics near ice-stream margins at the bed that are the primary cause of rib-formation?

**3    Minor/Editorial Points**

There are several of these but not enough to point out, given that a substantial revision will occur. The English is mostly excellent.

**4    References**

This is a list of all the papers I referred to in the review. Various works by the 'Fowler-school' (Chapwanya, Katz, Kyrke-Smith, Fannon) are cited in V++.

- Boulton, G.S., R.C.A. Hindmarsh, (1987), "Sediment deformation beneath glaciers: rheology and geological consequences", *J. Geophys. Res.*, **92B**(9), 9059-82

- Clark, C.D., (1993), "Mega-scale glacial lineations and cross-cutting ice flow landforms", *Earth Surface Processes and Landforms*, **18**, 1-29.

- Dunlop, P., C.D. Clark, (2006), 'The morphological characteristics of ribbed moraine', *Quat. Sci. Rev.*, **25**, 1668-91,

- Dunlop, P., C.D. Clark, R.C.A. Hindmarsh, (2008), "The Bed Ribbing Instability Explanation (BRIE) - testing a numerical model of ribbed moraine formation arising from coupled flow of ice and subglacial sediment", *J. Geophys. Res.*, **113**, F03005, doi:10.1029/2007JF000954.

- Eyles, N., (1993), "Earth's glacial record and its tectonic setting", *Earth Science Reviews*, 3**5**, 1-248

- Fowler, A.C., (2000), "An instability mechanism for drumlin formation", *Geological Society Special Publication*, **176**(1987), 307–319,

- Fowler, A.C., (2010), "Weertman, Lliboutry and the development of sliding theory", *J. Glaciol.*, **56**(200), 293-314

- Hart, J.K., R.C.A. Hindmarsh and G.S. Boulton, (1990), "Styles of subglacial glaciotectonic deformation in the context of the Anglian Ice Sheet", *Earth. Surf. Process. Landf.*, **15**(3), 227-41

- Hindmarsh, R.C.A., (1998a), "The Stability of a Viscous Till Sheet Coupled with Ice Flow, Considered at Wavelengths Less than the Ice Thickness", *J. Glaciol.*, **44**(147), 285-292.

- Hindmarsh, R.C.A., (1998b), "Drumlinization and Drumlin-Forming Instabilities: Viscous Till Mechanisms", *J. Glaciol.*, **44**(147), 965-72.

- Hindmarsh, R.C.A., (2004), " A numerical comparison of approximations to the Stokes equations used in ice-sheet and glacier modeling", *J. Geophys. Res.*, **109**(F01012)

- Kamb, B., C.F. Raymond, W.D. Harrison, H. Engelhardt, K.A. Echelmeyer, N. Humphrey, M.M. Brugman, T. Pfeffer, (1985), "Glacier Surge Mechanism: 1983-1983 Surge of Variegated Glacier, Alaska", *Science*, **227**(4686), 469-79

- Hättestrand, C., J. Kleman, (1999), "Ribbed moraine formation", *Quat. Sci. Rev.*, **18**(1), 43-61

- Röthlisberger, H., (1972), "Water Pressure in Intra- and Subglacial Channels", *J. Glaciol.*, **11**(62), 177-203

- Schoof, C., (2007), "Pressure-dependent viscosity and interfacial instability in coupled ice-sediment flow", *J. Fluid Mech.*, **570**(February), 227–252,

- Schoof, C., R.C.A. Hindmarsh, (2010), "Thin-film flows with wall slip: an asymptotic analysis of higher order glacier flow models", *Q. J. Mech. Appl. Math.*, **63**(1), 73-114

- Sergienko, O.V., R.C.A. Hindmarsh, (2013), "Regular patterns in frictional resistance of ice-stream beds seen by surface data inversion", *Science*, **342**, 1086-8

- Stokes, C.R., M. Margold, T.T. Creyts, (2016), 'Ribbed bedforms on palaeo-ice stream beds resemble regular patterns of basal shear stress ('traction ribs') inferred from modern ice streams', *J. Glaciol.*, **62**(234), 696–713

---

## Referee Comment (RC2) · Martin Ross (Referee) · 22 Feb 2021

This manuscript presents results from laboratory experiments, whereby bedforms are produced in fine sand under a small-scale 'ice sheet' analogue model made of silica gel using different water injection scenarios. The authors then compare results against real landform assemblages observed at three selected natural sites. One key aspect of these experiments is the ability of the gel to deform, in response to the water injection, in a way that produces a corridor of flowing gel that resembles an ice stream; bedforms in the sand are produced in the process. The authors thus make the case

that their analogue model is a scaled-down representation of an ice stream and that the landforms produced in the sand under the deforming silica gel are proportional and directly comparable to landforms associated to the ice stream landsystem. All the observed analogue landforms are described and mapped, but the analysis and interpretation focus on the oblique to transverse ridges, which are here considered analogous to ribbed moraines. The authors then find several similarities (e.g., morphometry, orientation, location) between the features produced by the analogue model and the features observed at the selected sites. The authors then suggest this finding has important implications on our understanding of ribbed bedforms in the specific context of ice streaming (i.e., how, and where they may form under ice streams), and that this work provides new criteria for palaeo-glaciological reconstructions of ice streams.

Analogue modelling is not new to the geosciences as it has been extensively used in the field of geodynamics to study folding and faulting, as well as to investigate larger scale problems in tectonics. To my knowledge, its application to ice sheet dynamics is quite novel, and based on the references cited, it was first applied just a few years ago to investigate tunnel valley formation. The experimental setup is interesting and seems to be done according to the state-of-the art. It would be useful to provide more details about scaling and how exactly the analogue model is similar in terms of geometry and dynamics to the natural system. The authors simply refer to earlier studies, but I think it is important given this is still relatively new to clearly explain scaling in this case (i.e., ice lobe/stream, landforms). It is also difficult to understand how a single lobe or stream is initiated from a central injection of water under a circular and uniform convex-up silicon cap. Why does it lead to a single lobe in one specific direction? I think an explanation about how the injection induces discrete deformation and movement along one specific direction in the silicon cap would be useful.

Analogue modelling is used for its advantages, which include 1- an ability to observe certain processes from beginning to end, 2- control the parameters to study different scenarios and, 3- record changes and map new features at any point in time during
the experiment. Analogue modelling can help scientists develop new ideas and help test some hypotheses. Therefore, I think they can be useful and the work presented in this manuscript is valuable and opens up new and interesting research avenues. However, any type of modelling is a simplification of the settings and processes that are taking place in real systems. Direct use of analogue modelling results to interpret natural phenomenon is a risky exercise and must be done with great caution. My general impression is that the authors make a direct link between the analogue and the natural cases too quickly, without properly explaining the main assumptions and the possible limitations of their model and of such comparison exercise. They do mention a few limitations later in the discussion (e.g. such as near line 555), but this should be more comprehensive and presented earlier. A full list of model simplifications and limitations, as well as assumptions for the comparison to natural phenomenon should be provided in the methods section. For instance, it seems the modelling ignores thermal effects. Lateral shear margins are not just wet/dry boundaries, but they are also thermal boundaries and several thermal/hydrological effects have been investigated and modelled (e.g. Haseloff et al. 2015, 2018; Meyer and Minchew 2018; Meyer et al. 2018).

Here is a list of more specific questions or problems that I think need to be addressed:

1) Four selected areas within only three paleo-ice streams represent a small sample to be confident about the degree of similarity between the analogue model and the real landsystems. The natural sites appear to have been based on a 'search and find features' strategy. I understand the rationale of doing this, but it introduces possible bias that may have an impact on the analysis and conclusion. This limitation should be acknowledged and discussed. It would be useful to identify some strategies to further assess the validity of the comparison exercise.

2) In the analogue model, ribbed bedforms developed obliquely to the flow direction near the lateral margins. Similar oblique features are also described such as near the lateral margin of the Amundsen Gulf ice stream. The authors cite previous work that

had interpreted these features as palimpsest glacial lineations; that is, older drumlins that got overprinted and ribbed following a shift in ice stream configuration. Considering the results of their analogue modeling experiments, the authors propose a new interpretation (see discussion near line 510), which is that these oblique lineations formed under the same ice stream configuration than for the younger glacial lineations that crosscut them. The crosscutting relationship between these features clearly indicate the oblique ribbed bedforms must have formed at an early stage. Furthermore, this two-stage process also brings the question of preservation. Are they observed near lateral margins because that is where they formed as oblique ribbed bedforms, or because older drumlins were better preserved there (i.e., only partially overprinted) due to lower flow velocities and patchy overprinting? It is an interesting idea to suggest they may have formed during a single phase (it would require at least a two-stage process) without any change in the configuration of the ice stream, but it remains to be tested. I would argue that there are more of these oblique ridges than shown on their figure 12b because the degree of overprinting and reworking increases toward the center of the trunk ice stream. So, they seem to have covered a wider portion of the bed than shown in Fig. 12b. Adding the water bodies could help visualize this better because some of the swales in-between the ridges have elongated lakes in them. The authors do recognize that some of these interpretations are preliminary and could be further tested (see lines between 530 and 533). Detailed ice flow reconstructions using independent measures like striations would help test these ideas. In summary, if the oblique bedforms formed in an earlier phase and were overprinted and drumlinized later by the streaming bed, their spatial distribution could reflect more the area of better preservation potential (erased more in the middle of the trunk than on the lateral edges). Are they ribbed bedforms from a single phase or palimpsest/overprinted drumlins turned into ribbed bedforms following a shift in ice stream configuration? I think the question remains open in my opinion.

3) Based on my above comments, I think it is premature to conclude that we now have new criteria for palaeoglaciological reconstructions.

4) The link to abrupt spatial variations in subglacial shear stress/basal drag/drainage has been proposed in previous models of ribbed moraine formation; perhaps not in the specific context of ice streams, but as a general process (i.e., ribbed bedforms develop under sticky areas of the bed in the presence of pre-existing till). The authors do recognize that ribbed bedforms and abrupt lateral and down-ice transitions with glacial lineations have been documented and interpreted to record large velocity gradients across the bed (near line 640). However, they say that these previous interpretations were only for very local sticky spots, which seems to suggest that they are of limited significance or that they could not apply to their case. I think that the ideas presented in this study are in many ways quite close to what was presented in these earlier publications. For me, this new study is interesting because it may provide a new way of testing these ideas. So, it is not so much a completely new explanation, but a new approach at testing previous interpretations, which does also seem to have the potential to bring new insights into processes. Another take home message is that there has been an emphasis on mapping and using flowsets in paleo-ice stream studies, but ribbed bedforms also provide key insights and thus deserve more attention because they can help understand the spatial patterns of sticky versus slippery portions of the bed, which is critical to understand ice stream dynamics and evolution.

5) This is more about the content and structure of the paper. I am wondering if the long section that reviews the paleo-ice stream landsystem is necessary. The rest of the paper focuses more on ribbed bedforms. Figure 1 is great; it is a high-quality conceptual model. So, I would suggest keeping that figure, but the text of section 2 could be considerably reduced. I think it would be sufficient to just summarize the conceptual model in the paper with appropriate references and use that space to present and discuss the assumptions, advantages, and limitations of analogue modelling for glacial dynamics problems. I think this is new/recent enough to justify more explanations.

This type of studies often brings more questions than answers, which is not a bad thing. It is hard to do innovative research, and I would thus like to commend the authors for

trying something like this. In summary, I think the results and the analyses are useful, promising, and will be of interest to the scientific community, but the authors must provide a complete list of assumptions and limitations and they also need to discuss interpretations and significance in a more balanced way.

Minor comments:

Line 34: Literature from the last 20 years is missing here.

Line 184: I agree with this. In most places I am familiar with, ribbed moraines tend to be variably overprinted/reworked, mostly by drumlins that have formed at a later stage (most cases).

Line 203: So, it is four processes; not three as listed above.

Line 214: I note this is the density of water at 20deg. C. Any implications for modelling ice-bed interface near the pressure-melting point?

Line 218: "...within the bed and along the silicon-bed interface"; in all directions or along one particular direction? Same question also for line 278...

Line 289: The feature is referred to as a 'delta' here, but it is not controlled by water level in a frontal water body. It would be more accurate I think to refer to it as a 'splay' or a fan.

Line 326: "similar"... Data would be useful here rather than having to rely only on visualization and terms like "similar".

Lines 515/16: That is excluding the possibility that on the lateral edges there could have been drumlins instead of MSGLs. I think without local information about ice flow phases (e.g. from striation data) the previous interpretation of palimpsest landscape cannot be eliminated.

Line 570: This is interesting. It would be important to link to the real case examples to support that statement.

Line 614-15: Yes, I agree with that.

Line 657: I would like to point out that long corridors or tracks of ribbed bedforms alternating with narrow corridors of drumlins occur in places like northern Manitoba and in mainland Nunavut. They form some kind of 'bar code' landscape (see Fig.1 and Fig.3 in Trommelen et al. 2014). The banding is probably too narrow and laterally repetitive to represent separate ice streams and ice stream margins, but it does suggest lateral and regular variation in basal stick-slip conditions.

Line 708: "development". Development or preservation? In real cases, they could be distributed like that because they were crosscut by channels. In other words, perhaps they were more widespread and laterally continuous in an early phase and then later crosscut/eroded by meltwater channels.

References: Haseloff, M. et al. (2015). A boundary layer model for ice stream margins. J Fluid Mechanics, 781: 353-387 Haseloff, M. et al. (2018) The role of subtemperate slip in thermally driven ice stream margin migration. The Cryosphere 12: 2545-2568 Meyer, C.R., Minchew, B.M. (2018). Temperate ice in the shear margins of the Antarctic Ice Sheet: Controlling processes and preliminary locations. Earth and Planetary Sci L, 498: 17-26 Meyer et al. (2018) A model for the downstream evolution of temperate ice and subglacial hydrology along ice stream shear margins, J Geophys Res, 123: 1682-1698

---

## Author Comment (AC1) · 15 Mar 2021

Reply to the referee's comments: Richard Hindmarsh

We thank Richard Hindmarsh for its very useful comments which have helped us to improve the quality and the precision of the manuscript. The major points raised by Richard Hindmarsh were regarding the discussion of processes occurring in the experimental model, with a focus on the physics involved in the formation of experimental bedforms compared to those of their natural counterparts. In section §2., the overview

of glacial landforms was reduced and refocus on ribbed bedforms descriptions and formations only. We would like to recall in this regard that the label "ribbed bedforms" used throughout the manuscript is purely descriptive and does not possess any implications in terms of formation processes. We did this to deliberately set a distance between the novel experimental observations we report and those in nature because uncertainty remains as to how closely connected they are. We suggest that some of the review comments have perhaps mistaken us to be saying that our experimental ribbed bedforms are ribbed moraines. In order to compare natural and experimental features, 'ribbed bedforms' gathers in a single label all the subglacial periodic ridges that are formed transverse or oblique to the ice flow direction with a ribbed appearance (excluding crevasse squeezed- ridges). Replies to referee's comments are addressed below in red. Annotations (1.) refer to paragraphs of review responses while annotations (§1.) refer to manuscript paragraphs.

Kind regards, Jean Vérité (on behalf of all co-authors)

1. "V++ rather steer clear of the viscous/plastic debate; they cite the ur-viscous paper (Boulton and Hindmarsh, 1987) but cite nothing by Terzaghi nor by Kamb." References regarding works of Terzaghi (1931), Kamb (1991) and Tulaczyk et al. (2000) on the plastic rheology of sub-glacial till were added in §2.3., which present the distinct formation processes, according to till rheologies, of ribbed bedforms.

2. a) "Their work does produce flow-transverse ribs, but does not produce features aligned with flow that resemble either drumlins or mega-scale glacial lineations. This raises the question of why do their experiments not produce the whole gamut of glacial geomorphology?" Except the hypothesis of "large-scale subglacial meltwater flood events" evoked by Shaw (2002), all the hypothesis regarding the formation of subglacial bedforms are associated to a till or a sedimentary substratum being coupled to an overlying ice sheet that flows and either deform and remould its bed. Streamlined bedforms (e.g. drumlins and MSGL), as opposed to transverse bedforms (e.g. ribbed bedforms), are mostly explained as formed beneath corridors of high ice flow velocity

(Clark, 1993; Stokes and Clark, 1999; Stokes et al., 2013). In our experiments, we inferred that ribbed bedforms are formed where the silicon cap and the bed are coupled (i.e. unlubricated interface) and where silicon flows $10^1$ times faster than the surrounding silicon cap. By comparison, silicon stream trunks – where ribbed bedforms do not form – exhibit velocities up to $3.10^1$ times faster than the surrounding silicon cap. The only way to trigger a fast-flowing corridor in the silicon model is to initiate decoupling above a pressurized water film. Where silicon flows the fastest (within the silicon stream trunk) the silicon-bed interface is thus entirely decoupled, therefore explaining the lack of drumlins and MSGL in our experiments. The maximum silicon flow velocity we experimentally reproduced in the silicon-bed coupling zones where ribbed bedforms develop is probably insufficient to initiate streamlined bedforms (see modifications in §5.2.2.). Indeed, the presumed velocities that enable drumlins and MSGLs formation are $10^1$ or $10^2$ times faster than where ribbed bedforms form, respectively (Stokes et al., 2013 - Fig.17).

b) "and brings us back to the question above about the physical realism of their simulations: are their laboratory ribs formed by the same set of processes that form subglacial ribs?" We agree that it is major point of the review to deepen the understanding of the formation process of experimental ribs and to determine if this process is compatible with those described in the literature. We now demonstrate that experimental ribbed bedforms are produced where the silicon cap is coupled to its underlying bed and the silicon undergone lateral or longitudinal velocity gradient because of heterogeneities in basal water drainage. The high basal shear stress generated by the coupled flow of silicon over the bed is responsible for bed deformation, characterized by the rotation and the boundary sliding of grain along intergranular shear planes (see the new Fig. 15 and modifications in §5.2.1.). The deformation of the bed initiates the formation of experimental ribbed bedforms and allows their growth. Experimental ribs thus form and elongate perpendicular to the shortening axes and parallel to stretching axes of silicon strain ellipses, and are consequently seen as controlled by silicon deformation. In this reviewed version, we decide to overfocus on the comparison of

shape and elongation of experimental ribs with superficial strain axes rather than flow directions (see modification of Figs. 7, 9, 14 and S2 and modifications throughout §4.). In the glacial literature four classes of formation processes are invoked to explain subglacial ribs (see §2.3.), and one is compatible with the process invoked to explain the formation of experimental ribs (see §5.2.2.). They consider (i) a deformable till layer – either with a plastic (Shaw, 1979; Bouchard, 1989; Lindén et al., 2008; Stokes et al., 2008) or a visco-plastic rheology (Hindmarsh, 1998a,b; Fowler, 2000; Schoof, 2007; Dunlop et al., 2008; Chapwanya et al., 2011; Fowler and Chapwanya, 2014; Fannon et al., 2017) –, flat and temperate, whose (ii) deformation results from the shear stress induced by the coupled flow of ice over an active till layer. The deformation of active till layers, in both rheological configurations, enables the formation of periodic subglacial ridges transverse and oblique to ice flow direction. As our experimental model has the same pre-requisite conditions (i and ii) and reproduces periodic ribbed bedforms, we consider that laboratory ribs can form by processes compatible with some existing theories of ribbed bedforms formation.

3. "The glacial geomorphology descriptions should be reduced substantially; I don't believe that, compared with their length, they contribute anything amazingly new. Rather, V++ should make the points that ribs, drumlins and MSGL exist, and V++ can simulate rib-formation." We agree that our inventory of glacial geomorphological was too exhaustive and too long regarding the main purpose of the paper: ribbed bedforms. Consequently, as a preamble of §2. we briefly introduce the different subglacial bedforms (ribs, drumlins, MSGL and hummocks) without detailed description, we deleted the sections dedicated to (i) streamlined bedforms, (ii) marginal and submarginal landforms and (iii) shear moraines. We focus §2. on ribbed bedforms and their morphological characteristics (§2.1.), their spatial distribution in glacial and ice stream landsystems (§2.2.) and their theories of formation (§2.3.).

4. "More attention to the basic physics is required; how did V++ estimate ice pressure, water pressure and effective pressure? From their plentiful quotations of theoretical

work by the viscous-till school it seems that they do not oppose this idea on fundamental grounds, so V++ should include more detailed analyses of how their observations of the component pressures relate to HFS." As mentioned in 2.b, the pre-requisite basal conditions ("(i) a deformable till layer, flat and temperate, whose (ii) deformation results from the shear stress induced by the overlying ice") and the bedforms reproduced in the laboratory support the viscoplastic-till theory, which consequently has to be physically tested. We totally agree that the distribution of effective pressure – potentially responsible for sediment flow and rib initiation – should be measured and monitored in order to better constrain the formation processes of experimental ribs. However, the technical limits of the experimental device do not allow to estimate the water and effective pressure at the scale of a single ribbed bedform. Twelve sensors measure the pore water pressure below the circular silicon cap, meaning we have two or three sensors below the silicon stream at most (see Fig. S3). Therefore, interpolating pressure maps at the stream scale would already be presumptuous and would not provide representative data at the bedform scale.

5. "My ideal paper form is [. . .] §2. - review of glacial geomorphological features produced under the ice, with an emphasis on rib descriptions (perhaps contrasting Rogen moraine and traction ribs) and on the viscous (and other) theories of rib formation; [. . .] ; §4. - a discussion of how the experimental set-up permits and/or disallows experimental observations that confirm existent theories." We thank the reviewer for this proposition and we agree with it. §2.: We agree with the remoulding of §2., with a more detailed and precise description of the distinct type of ribbed bedforms (a purely descriptive term gathering Rogen and ribbed moraines, traction ribs, mega-ribs and possibly other rib types yet to be described), notably regarding their wavelength and dimension compared with ice thickness. The different theories of ribbed bedform formation, their associated physics and the till rheology are now explicitly described. §4: As suggested by the reviewer, we added in the revised version of the manuscript new sections discussing the model and the experimental bedform formation. We discuss in this new section (§3.1.3.) the pros and cons of the experimental set-up in reproducing the subglacial environment processes, the subsilicon physical conditions and the silicon-water-bed interactions, notably regarding the subglacial processes we are able to reproduce or not. In §5.2.2., based on experimental results, we now discuss the compatibility of experimental ribbed bedforms with existent theories and lack of streamlined bedforms in our experimental landsystem.

6. "A newish feature of the analysis in V++ is the association of rib-field locations with particular locations in ice streams and ice sheets. This possibly leads V++ to overfocus on these - for example rib co-location with stream lateral margins and downstream ends of surge lobes - and ignore the widely agreed observation that rib fields are found upstream of drumlin fields, under slower-flowing portions of the ice sheet. Do V++'s results explain these?" Our experiments only produced ribbed bedforms in the locations we show and this is why we focus on these. Because the experiments did not produce drumlins or MSGLs, we cannot explain the ribbed moraine to drumlin transition. Also, we don't yet know whether the ribbed bedforms we describe in places across sharp velocity gradients (fast to slow) also occur across slow-to-fast gradients, and indeed whether they are even the same 'type of ribs' with the same process, or perhaps different things. Consequently, we agree that §2. had to be reshaped in order to overfocus on (i) the processes of ribbed bedforms formation, and on (ii) their spatial distribution, notably regarding their observed relationship with ice stream margins and velocity gradients. Although ribbed bedforms are more frequent below slow-mowing part of ice sheets, they have also been observed along palaeo-ice stream beds (e.g. Stokes et al., 2008, 2016; Moller, 2006, 2010; Stokes et al., 2016) and associated with regions of high velocity gradients between rather stagnant ice-sheet and fast-flowing ice (along borders of ice stream trunks or around subglacial sticky spots).

7. "The dependence of deformation rate on effective pressure (difference between the load exerted by ice and water pressure) was central to understanding how the instability arose; a particular point is that the dependence of flux on effective pressure affects the distribution of negative flux gradients (where the till is thickening). I would

like to see more consideration of how V++'s mechanisms of rib formation might be explained in terms of the Fowler-Schoof conditions." Comparing maps of silicon deformation rate and effective pressure, with maps of analog subglacial features would be fascinating and could potentially reconcile communities of palaeoglaciologists and glacio-physicians. However the current experimental device does not allow to produce high-resolution effective pressure maps (see 4. for more details).

8. "There is considerable focus in recent literature on the horizontal dimensions of the landforms compared with the thickness of the ice. [...] I recognise that in the past decade two 'species' of ribs have come to be recognised, the long-established ribbed/Rogen moraines with spacings of 300 - 1000 m ($\leq$ ice thickness), and the newer larger 'traction ribs' (Sergienko and Hindmarsh, 2013; Stokes et al., 2016) with spacings of a few kilometres ($\geq$ ice thickness); these traction ribs can be found underneath ice streams in non-traditional rib locations. [...] I encourage V++ in their resubmission to provide data on the rib spacings and how this compares with the silica gel thickness at time of rib formation" The wavelength of experimental lateral and submarginal ribbed bedforms lies in between 7 and 19 mm, and, 10 and 15 mm respectively. The rib wavelength tends to remain constant once lobe stabilizes and thinning rates of the silicon layer reduces (i.e. as the lobe advances, by conservation of mass, the stream becomes thinner). For a stabilized lobe, the profile of silicon thickness slowly decreases up to the margin and the column of silicon is closed to 20 mm above the lateral ribs (i.e. formed below shear margins) and between 5 and 15 mm above the submarginal ribs (i.e. formed below lobes). The experimental rib wavelength lies in between 0.4 and 1.5 silicon thickness, and thus tends to form with a spacing slightly lower than or equal to the cap thickness. Experimental rib characteristics compare quite well with traction ribs whose spacing is equal to or greater than ice thickness (Sergienko and Hindmarsh, 2013; Stokes et al., 2016), while ribbed and Rogen moraines exhibit a spacing (Dunlop and Clark, 2006) almost an order of magnitude less compared with ice thickness (Paterson, 1972). This new morphometric argument in favour of experimental ribs similar to traction ribs is now added in §5.1. However, to establish a semi-quantitative law

comparing the evolutions of bedform wavelength and cap thickness from experiment model would not be suitable since the silicon-bed system is physically different from the ice-till system.

9. "If the horizontal dimension is less than or comparable with the ice thickness, the full Stokes equations need to be solved in order to calculate the normal stress exerted by the ice on the sediment accurately; this is needed to calculate the effective pressure. [. . .] They could also comment on how their observations coordinate with the Fowler-Schoof conditions for geomorphological instabilities to exist. [. . .] It is almost certain that modelling traction ribs requires solution of the Stokes equations, despite their large horizontal dimensions, owing to the slippery beds (low 'traction number') beneath streams (Hindmarsh, 2004; Schoof and Hindmarsh, 2010)." Although experimental ribs display horizontal dimensions similar to traction ribs, technical limitations of the experimental device do not allow to solve Stokes equations and to test Fowler-Schoof conditions. With the resolution of current pressure measurement, we are not able to calculate effective pressure at the scale of ribbed bedforms. Despite this technical inability to quantitatively/physically test theories of rib formation, our experimental model enables us to discuss the distribution, the orientation and the formation timing of ribs compared to ice and meltwater flows; and thus to discuss similarities between our experimental model and existing theories.

10. "A substantial proportion of theories of rib-formation (e.g. Hättestrand and Kleman, 1999) focus on the freezing-melting boundary map-location as a control on rib formation. V++'s experimental set-up does not permit investigation of this aspect, but this matter does require some comment, in particular on the issue of whether, subglacially, there is one and only one means of forming ribs." Experiments are realized in a laboratory whose temperature lies in between 15 and 20°C that consequently do not reproduce frozen basal conditions. An experimental frozen-bed will also limit the formation of bedforms by increasing bed stiffness and cohesion, and reducing the bed deformability and erodibility. One theory of rib formation, established by Hättestrand

and Kleman (1999), suggests that transition in the basal thermal regime could induce frozen bed fracturing and cause the formation of ribbed bedforms. As mentioned in 2.b, this theory is not testable with our experimental device since the bed is temperate. Others theories however exist and ribbed bedforms might be formed through several means (§2.3.), one of which invoked bed deformation in response to the coupled flow of ice, water over a deformable till layer and is compatible with our model (§5.2.2.).

11. "Were 'faults' observed, and what does this tell us about the styles of deformation?" No faults are observed within the sedimentary bed at the scale of experimental bedforms. The mean height (hmean) of analog ribbed bedforms is equal to more or less 1mm while the median grain size of the bed is dmed = $100\mu$m (i.e. a rib is composed of up to 10 sand grains). Although it might be very hard to distinguish any structures in homogeneous sand bed considering the size of the ribbed bedforms, cross-section making are very difficult to make because they tend to destroy the structures due to pressure exerted on the water-saturated bed and the silicon during the realisation of slices through the model. Given the bed is constituted of unconsolidated and water-saturated sand with very low cohesion, and almost equant grains lack a clear lattice-preferred orientation, we assume that the bed undergoes ductile deformation through transfers of sediments at grain-scale through grain boundary sliding, translation and rotation possibly along intergranular shear planes in the upper active bed (Hamilton et al., 1968; Oda and Konishi, 1974; Owen, 1987; Bestmann and Prior, 2003). Considering the size of our experimental ribbed bedforms, these intergranular shear planes might correspond to "faults" in nature (see modifications in §5.2.1. and the new Figure 14).

12. "Some thought needs to be put into explaining why the experiments do not produce flowaligned features (drumlins/MSGL) in the context of work by the Fowler-school modelling of drumlin formation. I appreciate that a definitive answer may not be available yet, but this would be of considerable benefit to those wishing to extend and elaborate the work of V++." Whatever the till rheology, viscoplastic (Hindmarsh, 1998a,b; Fowler,

2000; Schoof, 2007) or plastic (Terzaghi, 1931; Kamb, 1991; Tulaczyk et al., 2000), the ice flow transmits a shear stress to an active till layer when the ice is coupled to its bed, thus triggering bed deformation and bedform initiation (ribs, drumlins and MS-GLs). The only difference being that numerical models considering a viscoplastic till rheology are able to quantitatively predict bedform spacing and produce timedependent bedform evolution (Fannon et al., 2017). Regardless of the till behaviour, it is assumed that mega-scale glacial lineations are thought to be formed under high flow velocity or longer duration than drumlins, and even higher than ribs. Some studies suggest that ice velocity is a key parameter considering the modelled basal velocities and the improbability of constant and straight ice flow for hundreds of years during deglaciation periods (Clark, 1993; Stokes et al., 2013). Based on these arguments and on those enounced in 2.a about the experimental stream velocity, it is probable that silicon velocities are insufficient to initiate streamlined bedforms or to generate ribbed bedforms evolving into drumlins, while the fundamental basal processes are met (see modifications in §5.2.2.).

13. "Since glacier linked-cavities rely on cavity formation and hydraulic links between the cavities, V++'s association of laboratory-observed networks with 'linked cavities' is quite reasonable, but R-channel theory relies heavily on the heat production by flowing water melting the tunnel that is being closed by the weight of the ice; thermal effects are not included in V++'s experiments. A related point is that R-channel theory and linked-cavity theory have opposite relations between the system transmissibility (product of permeability and vertical area) and effective pressure; Rchannels have transmissibility increasing with effective pressure, while linked cavities have it decreasing. It is not clear whether the dynamics of the sub-silica drainage system are the same as the sub-glacial; for example, might not the lab drainage system development be due to a Hele-Shaw instability? It might be that V++ wish to point out the similarities between the mechanisms of their lab-formed streams and streams in the field, but the real question is whether sufficient observations have been made in either case; I'm pretty sure that not enough is known about stream-formation in nature." Sub-silicon drainage system is composed of three type of drainage features: (i) water film corresponding to the distributed and widespread flow of water below the experimental ice stream (Breemer et al., 2002; Le Brocq et al., 2009); (ii) linked cavities corresponding to interconnected water spots in between ribbed bedforms (Fowler, 1987); and (iii) N-channels (rather than R-channel) corresponding to meltwater channels and tunnel valleys carved in the sedimentary bed (Madsen, 1921; Nye, 1973). As the silicon putty cannot reproduce the ice–water phase transition and the temperature does not affect silicon putty properties, sub-silicon water flow cannot generate Rchannels. Hele-Shaw instabilities can occur at the interface of two fluids (of distinct viscosities as water and silicon putty) flowing at different velocities, and generate periodic wave-like features transverse to the interface. It is therefore unlikely that this instability is responsible for the formation of the drainage system. In our experiment, N-channels develop in response to the erosive flow of pressurized water that carves channelized features into the bed (Lelandais et al., 2016). They form in order to improve the effectiveness of the water drainage system, draining water from the highpressure water film to the low-pressure marginal area. Those mechanisms are similar to processes responsible for meltwater channelization in subglacial environments (Fountain and Walder, 1998; Kehew et al., 2012; Lelandais et al., 2016; 2018). Experimental linked-cavities develop when the bed surface becomes no longer flat and bedforms grow. Water from the water film spreads in between those topographic obstacles, notably in the lee-side of ribbed bedforms. As experimental ribs are organised into fields, cavities are interconnected and constitute a water drainage system at the silicon stream margin similar to subglacial linked-cavities (Fowler, 1987; Hooke, 1989).

14. "V++ do not include thermally-based mechanisms in their experiments, which leads naturally to wondering about their lab-produced ribs adjacent to stream boundaries is this saying (as they seem to be suggesting) that one condition for rib formation is a large lateral velocity gradient - a glaciological insight of potentially great importance - or are there some special thermal characteristics near ice-stream margins at the bed that are the primary cause of rib-formation?" A consequence of thermal mechanisms along

the basal interface of glaciers is the subglacial production of meltwater able to initiate, in certain cases, decoupling between the ice and its bed. In areas where the bed is permeable or where meltwater production is absent, ice-bed coupling predominantly occurs. In the experiments, the spatial transition in meltwater availability and drainage types between mostly coupled areas (e.g. lateral shear margins and lobe margins) and decoupled areas (e.g. silicon stream) induce flow velocity gradients. Beneath the silicon shear margins, lateral velocity gradients between the silicon stream and the stagnant surrounding cap generate high basal shear stress being accommodated through bed deformation and ribbed bedforms formation. Ribbed bedforms formation in our experiments is thus primary conditioned by the transition of the basal meltwater drainage (not reproduced by thermal variations in the model but by injection of water beneath the silicon; see §3.1.3.), which induce velocity gradients delineating the ice stream margin.

---

## Author Comment (AC2) · 15 Mar 2021

Reply to the referee's comments: Martin Ross

We thank Martin Ross for their comments which have helped us to improve the manuscript. Replies to referee's comments are addressed below in red. Annotations (§1.) refer to the corrected manuscript paragraphs. Kind regards, Jean Vérité (on behalf of all co-authors)
Major comments

1. "It would be useful to provide more details about scaling and how exactly the analogue model is similar in terms of geometry and dynamics to the natural system. The authors simply refer to earlier studies, but I think it is important given this is still relatively new to clearly explain scaling in this case (i.e., ice lobe/stream, landforms)."

Our experimental device primarily involves water flow, within the bed and along the basal interface either channelized or distributed. This water flow controls both the silicon cap dynamics, responsible for bed deformation and sedimentary processes (erosion, transport and deposition). Thus, the scaling has to take into account the complex relation between silicon flow, water flow and subglacial landform development. As the experimental materials were selected in order to favour the reproduction of some key processes between the natural system and the experimental model (e.g. basal decoupling, bed erodibility and deformation) and to provide critical advantages lacking in the natural system (e.g. transparency to observe the interface and low viscosity of the cap to observe significant flow at the experiment timescale), some ratios of physical parameters (e.g. viscosity) are different, excluding a perfect scaling. However, as described in the section §3.1.1., Lelandais et al. (2016, 2018) proposed a scaling for a reduced-size version of our experiment model, considering the silicon lobe dynamic, the water flow and the subsilicon landform development. They demonstrate that the dimensionless ratio between the lobe margin velocity and the incision rate of subglacial erosional landforms has similar values in the model and in nature. In the same way, we demonstrate in this study that the dimensionless ratio between the ribbed bedform wavelength and the overlying cap thickness display compatible values in the model and in the nature (see §5.1.). Considering we aim to focus on relations between basal water flow, cap dynamic, bed erosion and deformation, we assume that these comparable dimensionless ratio between nature and experiments are satisfying. Furthermore, we now present in the new section §3.1.3. what the experimental device does not reproduce (e.g., shear heating, heat softening, melting, freezing, crevassing) and which elements

are not scaled in the experimental device (e.g., shear margin width, channelized feature width).

2. "Why does it lead to a single lobe in one specific direction? I think an explanation about how the injection induces discrete deformation and movement along one specific direction in the silicon cap would be useful."

In our experiments, the emergence of a system of fast-flowing stream and lobe is controlled by the growth and migration of a water pocket. The water pocket – whose formation results from the injection of water beneath the silicon – migrates toward the margin of the silicon cap following the pressure gradient. The pressure gradient is potentially influenced by the distance between the injection point and the silicon margin, small variations in bed permeability, variations in bed or silicon surface slopes. Although we aim to ensure a constant experimental protocol with almost identical input parameters, small variations in the above parameters can occur during experiment preparation. These variations will therefore control the pressure gradient and hence the migration path of water pocket and the formation of the silicon stream and lobe system. The specific direction of lobe development is mostly unpredictable.

3. "They do mention a few limitations later in the discussion (e.g. such as near line 555), but this should be more comprehensive and presented earlier. A full list of model simplifications and limitations, as well as assumptions for the comparison to natural phenomenon should be provided in the methods section. For instance, it seems the modelling ignores thermal effects."

We agree that an exhaustive description of model physics, simplifications and limitations is necessary and we added a new section dedicated to this point in the revised manuscript (§3.1.3.). Furthermore, we complete the section dedicated to the discussion of experimental ribbed bedforms with their natural counterparts (§5.2.2.), notably regarding their formation processes. Those two sections thus present a comprehensive and clear discussion of what it is reproduced in the experimental model and how it

fits with natural processes, which makes it possible to reasonably discuss the meaning of ribbed bedforms in glacial landsystem based on our experimental results (§5.3.).

4. "The natural sites appear to have been based on a 'search and find features' strategy. I understand the rationale of doing this, but it introduces possible bias that may have an impact on the analysis and conclusion. This limitation should be acknowledged and discussed. It would be useful to identify some strategies to further assess the validity of the comparison exercise."

We agree that the mapping of ribbed bedforms along ice stream margins has been based on a 'search and find features' and that the small sample questions on the statistical representativeness of our observations. In this study, the aim is to demonstrate that ribbed bedforms form with characteristic shapes below lateral shear margins and frontal lobes of ice streams, as the experiments suggest it. We illustrate this suggestion through a selection of four sections of palaeo-ice stream beds, even other natural examples of oblique ribbed bedforms bordering lateral ice stream margins occur in the south of Wollaston Peninsula (69°12'N; 111°55'W), in the northeast of Ireland (54°1'N; 7°28'W) and in several sectors of the Scandinavian Ice Sheet (Szuman et al., 2021) for examples. Despite those observations, we agree that the kind of ribbed bedforms we present in this study were probably not systematically formed or preserved below lateral shear margins and frontal lobes of palaeo-ice streams, what must be highlighted in the discussion (see modifications in §5.1.). Consequently, several future investigations have to be realized in order to test the hypothesis that ribbed bedforms could be a morphological criterion to identify large-scale lateral or longitudinal velocity gradients (e.g., lateral shear margins and frontal lobes of ice streams). We suggest to (i) investigate a larger sample of palaeo-ice stream beds in order to explore the conditions of preservation and formation of these characteristic ribbed bedforms along their margins (see §5.1.) and (ii) prospect large-scale corridors and belts of ribbed bedforms in order to search if they could fit with hitherto non-recognized palaeo-ice streams (see §5.3.1.).

5. "In the analogue model, ribbed bedforms developed obliquely to the flow direction

near the lateral margins. Similar oblique features are also described such as near the lateral margin of the Amundsen Gulf ice stream. [. . .] Considering the results of their analogue modelling experiments, the authors propose a new interpretation [. . .], which is that these oblique lineations formed under the same ice stream configuration than for the younger glacial lineations that crosscut them. The crosscutting relationship between these features clearly indicate the oblique ribbed bedforms must have formed at an early stage. Furthermore, this two-stage process also brings the question of preservation. Are they observed near lateral margins because that is where they formed as oblique ribbed bedforms, or because older drumlins were better preserved there (i.e., only partially overprinted) due to lower flow velocities and patchy overprinting? It is an interesting idea to suggest they may have formed during a single phase (it would require at least a two-stage process) without any change in the configuration of the ice stream, but it remains to be tested. [. . .] In summary, if the oblique bedforms formed in an earlier phase and were overprinted and drumlinized later by the streaming bed, their spatial distribution could reflect more the area of better preservation potential (erased more in the middle of the trunk than on the lateral edges). Are they ribbed bedforms from a single phase or palimpsest/overprinted drumlins turned into ribbed bedforms following a shift in ice stream configuration? I think the question remains open in my opinion."

The bed of the Amundsen Gulf Ice Stream (AGIS) display a complex assemblage of two symmetric fields of oblique and elongated bedforms along both lateral margins of a trunk, characterized by a swarm of streamlined bedforms parallel to the trunk axis. The internal part (i.e., close to the trunk axis) of oblique bedform fields is partially overprinted by lineations with an orientation identical to the streamlined bedforms observable along the trunk. Those morphological observations fed two hypotheses in the glacial literature. Winsborrow et al. (2004) and Stokes et al. (2006), interpreted those oblique and crosscutting bedforms as three swarms of streamlined bedforms – preserved, partially overprinted or fully eroded – associated to different flowsets. Greenwood and Kleman (2010) think the same way, except that they interpret the

oblique and elongated bedforms occurring along the margins and partially overprinted by lineations as mega-ribs rather than streamlined bedforms. Based on experimental data, we suggest that oblique ribbed bedforms can form below lateral shear margins of ice streams, areas experiencing large-scale lateral velocity gradients, and develop peripheral to swarms of streamlined bedforms. We therefore propose that oblique and elongated bedforms described in the AGIS resemble in shape and pattern to experimental oblique ribbed bedforms and are potentially the same, with the difference that ribbed bedforms beneath the AGIS are apparently overprinted by lineations. This kind of oblique ribbed bedforms overprinted by streamlined features resembles (i) ribbed moraines with superimposed drumlins, both illustrating a single flowset (Greenwood and Clark, 2008), (ii) transverse asymmetrical drumlins (Shaw, 1983) and (iii) drumlins with an "en Echelon" arrangement (Clark, 2018). In parallel, the idea of a continuum in subglacial bedforms (ribbed bedforms, drumlins and MSGLs) emerged in recent modelling (Fowler and Chapwanya et al., 2014; Barchyn et al., 2016; Fannon et al., 2017) and palaeo-glaciological works (Stokes et al., 2013; Ely et al., 2016). Given that, it seems reasonable to propose a third hypothesis considering that drumlinized and oblique ribbed bedforms are an intermediate bedform between ribbed bedforms and drumlins. In this way, streamlined bedforms, and, oblique and overprinted ribbed bedforms can co-exist and form beneath a single ice stream undergoing spatial and temporal variations in flow velocity. Currently we cannot demonstrate this hypothesis but it needs to be considered. We clarify this discussion regarding the bedforms observable along the bed of the Amundsen Gulf Ice Stream and clearly enounce that the significance of oblique and drumlinized ribbed bedforms remains an open question with distinct hypotheses (see modifications in §5.1.).

6. "Based on my above comments, I think it is premature to conclude that we now have new criteria for palaeo-glaciological reconstructions."

We agree that our suggestion to interpret specific kind of ribbed bedforms as an additional morphological criteria to identify palaeo-ice stream margins is based on a reduced natural sample (Major comment 4.) and in certain cases on a new interpretation of bedforms (Major comment 5.). Even if a future overview of palaeo-ice stream margins will be necessary in order to validate this hypothesis, with regards to experimental and natural results presented in this study, we believe that it is justified to propose that the ribbed bedforms can constitute a new and supplementary criteria for palaeo-glaciological reconstructions, especially in identifying ice stream margins In the revised version of the manuscript, we keep and discuss this hypothesis (see modifications in §5.3.1.).

7. "The authors do recognize that ribbed bedforms and abrupt lateral and down-ice transitions with glacial lineations have been documented and interpreted to record large velocity gradients across the bed (near line 640). However, they say that these previous interpretations were only for very local sticky spots, which seems to suggest that they are of limited significance or that they could not apply to their case. I think that the ideas presented in this study are in many ways quite close to what was presented in these earlier publications. For me, this new study is interesting because it may provide a new way of testing these ideas. [. . .] There has been an emphasis on mapping and using flowsets in palaeo-ice stream studies, but ribbed bed forms also provide key insights and thus deserve more attention because they can help understand the spatial patterns of sticky versus slippery portions of the bed, which is critical to understand ice stream dynamics and evolution."

It is now a widespread idea that ribbed bedforms form were ice slows down either at short (i.e., sticky spots; Stokes et al., 2007, 2016) or large scale (i.e., onset area of ice stream and ice dome; Aylsworth and Shilts, 1989; Dyke et al., 1992; Hättestrand and Kleman, 1999; Greenwood and Kleman, 2010; Stokes, 2018). Consequently, the formation of ribbed bedforms thus provide some key information regarding the ice velocity pattern and the spatial variations of sticky and slippery portions of bed, controlled by basal thermal regime, bed rheology and hydrological conditions. Experimental results confirm this idea and demonstrate that ribbed bedforms with specific shapes and orientations develop below lateral shear margins and frontal lobes of ice streams, where lateral and longitudinal velocity gradients occur in response to variable hydrological conditions along the basal interface. We agree that the conditions responsible for the formation of ribbed bedforms both in local sticky spots and along ice stream margins must not be opposed. Indeed those conditions confirm that the formation of ribbed bedforms is controlled by the spatial pattern of sticky versus slippery portions of the bed, whatever their scale. We rectify this point in the revised version of the manuscript (see §5.3.1.).

8. "I am wondering if the long section that reviews the palaeo-ice stream landsystem is necessary. The rest of the paper focuses more on ribbed bedforms. [. . .] The text of section 2 could be considerably reduced. I think it would be sufficient to just summarize the conceptual model in the paper with appropriate references and use that space to present and discuss the assumptions, advantages, and limitations of analogue modelling for glacial dynamics problems."

We agree that our review section of palaeo-ice stream landsystem is long compared with the main purpose of the paper, the ribbed bedforms. Consequently, we reduce the state-of-art section (§2.) and focus on the ribbed bedforms, their shape, their spatial distribution and their formation processes. We also add a new section discussing the advantages, the limitations and the physics of silicon-water-bed interactions in the experimental model (§3.1.3.).

Minor comments

Line 34: Literature from the last 20 years is missing here. References "Kyrke-Smith et al., 2014" and "Minchew & Joughin, 2020" are added in the revised version of the manuscript (see §1.)

Line 184: I agree with this. In most places I am familiar with, ribbed moraines tend to be variably overprinted/reworked, mostly by drumlins that have formed at a later stage (most cases). "Moreover, ribbed bedforms are frequently overprinted by drumlins and

embedded within polygenetic landsystems, corresponding to multiphase stories, which complicates their interpretation" (see §2.3.)

Line 203: So, it is four processes; not three as listed above. We corrected this error (see §2.3.)

Line 214: I note this is the density of water at 20deg. C. Any implications for modelling ice-bed interface near the pressure-melting point? Modelling experiments are realized in a laboratory with constant temperatures lying in between 15 and 20°C. We consequently do not reproduce the effects of temperature dependence on the silicon-water-bed system, as frozen basal conditions and melting processes. It constitutes a limitation of our experimental model, now mentioned in the dedicated §3.1.3. section of the revised manuscript.

Line 218: "... within the bed and along the silicon-bed interface"; in all directions or along one particular direction? Same question also for line 278... A fraction of the injected water (75%, see Lelandais et al., 2018 – Section 2.1) flows within the permeable bed in all directions from the water injector. The discharge of injected water is calculated beforehand so that water pressure exceeds the combined weight of the sand and silicon layers, and to allow the flow of water at the silicon–substratum interface. Water at the silicon-bed interface first accumulates above the water injector and forms the water pocket. Once a preferential direction for water routing is established during the water pocket migration (see Major comment 2.), the water flows in the same specific direction during the rest of experiment.

Line 289: The feature is referred to as a 'delta' here, but it is not controlled by water level in a frontal water body. It would be more accurate I think to refer to it as a 'splay' or a fan. First, all sediments deposited at the exit of sub-silicon conduits are deposited beneath water, indeed water accumulates around the silicon cap through the experiments as we keep injecting water. We partly agree with your comment above. In the very first stage of channelization and tunnel valleys development in the experiments, subaqueous fan

in contact with the silicon forms, at this time sediment deposition is not controlled by water level and no stream-deposited topset is observed (see ice-contact fan definition by Lønne, 1995). However the fan builds up quickly to the water-level and evolve into an ice-contact delta where channelized stream deposits form topsets at the top of the delta in its proximal area (see ice-contact delta definition by Lønne, 1995). We made sure that this distinction is clearly written in the revised version of the manuscript.

Line 326: "similar" ... Data would be useful here rather than having to rely only on visualization and terms like "similar". The similarity between experimental and natural ribbed bedforms is based on qualitative data (bedforms with an undulating crest that form transverse to oblique to the flow direction) and quantitative data (elongation ratio, l/w = 3-4, and periodicity, with a wavelength slightly higher than their width). Even those data are already described both in sections §2.1. and §4.1.1. for natural and experimental ribbed bedforms respectively, we clearly write those data in order to argument this comparison in the revised version of the manuscript.

Lines 515/16: That is excluding the possibility that on the lateral edges there could have been drumlins instead of MSGLs. I think without local information about ice flow phases (e.g. from striation data) the previous interpretation of palimpsest landscape cannot be eliminated. See the response to Major comment 5. for more details.

Line 570: This is interesting. It would be important to link to the real case examples to support that statement. The ribbed bedforms mapped in this study along the lateral shear margins of the Amundsen Gulf Ice Stream, the Hay River Ice Stream and the Central Alberta Ice Stream display elongation ratio of 4.2, 2.9 and 3.8 respectively. The comparison classic ribbed/Rogen moraines (elongation ratio = 2.5; Dunlop and Clark, 2006) confirm our suggestion that real cases examples of ribbed bedforms observed below lateral shear margins present high elongation ratio because of the specific stress condition under which they form.

Line 657: I would like to point out that long corridors or tracks of ribbed bedforms alternating with narrow corridors of drumlins occur in places like northern Manitoba and in mainland Nunavut. They form some kind of 'bar code' landscape (see Fig.1 and Fig.3 in Trommelen et al. 2014). The banding is probably too narrow and laterally repetitive to represent separate ice streams and ice stream margins, but it does suggest lateral and regular variation in basal stick-slip conditions. We agree that the observation of ribbed bedforms clustered into corridors and belts, similar to belts reproduced it in the analog model, should not be interpreted restrictively as potential margins of palaeo-ice stream. As mentioned in our response to the Major comment 7., the formation of ribbed bedforms is controlled by the spatial distribution of sticky and slippery bed. Consequently, a future detailed mapping of long parallel corridors alternating with ribbed bedforms and lineations, as it is observed in northern Manitoba and near the Lake Naococane for examples, could provide information regarding the lateral and regular variations in basal stick-slip conditions. Based on our experimental results, corridors of oblique ribbed bedforms could potentially illustrate bands with shear deformation resulting from lateral velocity gradient. This point is now clearly nuanced in the section §5.3.1. of the revised manuscript. We are also more exhaustive regarding the ice dynamics and basal condition information that long corridors of oblique ribbed bedforms could provide.

Line 708: "development". Development or preservation? In real cases, they could be distributed like that because they were crosscut by channels. In other words, perhaps they were more widespread and laterally continuous in an early phase and then later crosscut/eroded by meltwater channels. We agree that some ribbed bedforms are crosscut by channels and were more widespread and continuous before this crosscutting/erosion phase. However, one interpretation made in this study – based on experimental and natural observations – is that formation of channelized features below a frontal lobe generates longitudinal shortening stress along the coupled basal interface in between channels and triggers the development of submarginal ribbed bedforms, resulting from compressive bed deformation. Even if the channelization of water below lobes erodes submarginal ribbed bedforms and preserves them in between the asso-

ciated channels, it firstly allow the formation of ribbed bedforms. That's why we keep the term "development" in the revised version of the manuscript.

---

## Referee Report (RR1)

**Review of "Ribbed bedforms in palaeo-ice streams reveal shear margin positions, lobe shutdown and the interaction of meltwater drainage and ice velocity patterns" by Jean Vérité and ten others for *The Cryosphere**

Richard C.A. Hindmarsh

British Antarctic Survey and University of Durham

May 25, 2021

**1 Overview**

This revised paper by Jean Vérité and colleagues (V++) from *Le Mans Université*, *Université de Nantes*, University of Sheffield and Alberta Geological Survey addresses, as did the first submission, a centuries-old topic in glacial geomorphology, namely the mechanisms operating and conditions needed to produce certain glacial geomorphological landforms; in particular their focus is on ribs transverse and obliquely transverse to the ice flow direction. They discuss these ribs within the context of ice-streams; a key feature of the revision is provided by the sentences "*Within this general model, the meaning of ribbed bedforms remains unclear. Different types and shapes have been described, different theoretical models have been proposed for their formation, various interpretations exist for their timing and conditions of formation compared to ice and subglacial drainage dynamics, and their spatial distribution within glacial landsystems is still debated*" (l.81-84). This helps enormously; previously I had difficulty in ascertaining the function of the paper; now I see that it is aimed at the spatial distribution of sub-glacial ribs, and emphasises that they do not only occur under slower moving parts of the ice, but also in stream-marginal locations of various kinds - I mentioned in my first review that they wish to focus on some previously underconsidered aspects of geomorphology.

The lab experiments produce rib forms in what glacial geomorphologists might regard as unusual locations, and V++ wish to emphasise the spatial realism of their results.

V++ use an integrated approach, simulating the ice sheet with a pancake of flowing silica gel, which overlies sand, and pump water in underneath in order to lubricate the base. As part of the experiments, silica streams form, which V++ compare with ice streams.

The paper is organised into six main sections, containing sub-sections and sub-sub-sections; which bring out the logical flow of the paper;

- §1 *Introduction*: A one-page focus on the landsystems associated with former ice-streams

- §2 *Ribbed bedforms in ice stream landsystems*: Contains

    §2 Introductory subsection: Includes key quote referenced above, commencing "*Within this general model, the meaning of ribbed bedforms remains unclear...*"

    §2.1 *Types and shapes*: Emphasises that V++ use "*ribbed bedforms*" as a descriptive term with no genetic implications.

    §2.2 *Spatial distribution in ice stream landsystems*: Indicates that not only are ribbed bedforms created under slower-moving areas of ice (a view current in 2021) but also in association with streamlined forms e.g. drumlins, a view underemphasised at the moment.

    §2.3 *Theoretical models of formation*: A brief one paragraph review of current models of ribbed bedform genesis.

- §3 *Methods*

    §3.1 *Analog modelling*: Describes lab setup and experiments.

    §3.2 *Mapping and morphometric analysis on palaeo-ice stream landsystems*: Outlines V++'s work on mapping landforms in selected areas.

- §4 *Results*

    §4.1 *Ice stream dynamics and development of ribbed bedforms in the experiment*: Discussion of two stream-types found in the lab experiments and how they relate to evolution of bedform fields.

    §4.2 *Palaeoglaciology: ribbed bedforms beneath ice stream margins*: Focus on ribbed patterns close to stream lateral and downstream margins (this is a novel to the palaeo-glaciological community feature of V++'s work).

- §5 *Discussion*:

§5.1 *Morphometric comparisons between experimental and natural ribbed bedforms*: Compares lab results with an extensive set of field observations.

§5.2 *Processes of ribbed bedform formation at ice stream margins*: Focus on processes occurring at stream margins in the lab and in the field.

§5.3 *Ribbed bedforms in ice stream landsystems*: Emphasises that both lab and field rib formation not only occur not only in the slower-moving areas but also in special marginal areas of the fast-flowing streams.

- §6 'Conclusion': Key sentence "*Despite the ubiquitous and extensive nature of subglacially-produced ribbed bedforms beneath former ice sheets (variously called Rogen moraine, ribbed moraine, ribbed bedforms, mega-scale transverse ridges), their significance in ice stream landsystems and their formation processes remain poorly understood*"; this is a major point arising from V++'s work.

My summary of the contents indicates that V++ have made a serious effort to revise the paper in accordance with the referees' requests, and that the logical flow is much more straightforward; the first submission gave the impression of trying to solve *all* the problems of glacial sedimentary geology. In my first review I said that "V++ presents novel exciting experimental work which will inspire a very large portion of the glaciological community, and should certainly be published" - I believe it to be publishable now, and recognise that any differences between me and the authors are essentially 'matters of opinion', and that research over the coming decade(s) will clarify most of the issues.

The paper contains two key points; (i) rib formation is not confined to slower moving areas of ice; and (ii) there are distinct analogies between the slow viscous flows of ice-sheets over clastic sediments, and the slow viscous flow of silica over sand.

I am happy for the paper to be published; I don't have any 'Major Points' to be dealt with and encourage the authors to act on the minor points below.

**2   Minor/Editorial Points**

TThe English is mostly excellent, but ...

1. l.118-119. Sentence "*Ribbed bedforms are thus believed to be ubiquitous from the inner, cold-based, regions of ice sheets*". This sentence seems to saying that bedforms are created under frozen bed conditions - is this what V++ intend? Perhaps change to "inner, slower-moving, regions of ice-sheets"?

2. l.155. "*3.1 Analog modelling*" change to "*3.1 Analogue modelling*"; 'analog' is US-based spelling, *The Cryosphere* is a European-based journal.

3. l.264 "*4.1. Ice stream dynamics and development of ribbed bedforms in the experiment*". Is the focus in this section on ice-stream dynamics or silica-stream dynamics? Perhaps change to "*4.1. Stream dynamics and development of ribbed bedforms in the experiment*".

4. Nearly everywhere: change "*silicon*" to "*silica*"; 'silicon' is an element, 'silica' is a silicon-based substance used in the experiments.

---

## Editor Decision (ED1)

[revised manuscript text omitted]

**2.3. Theoretical models of formation**

Beneath warm-based ice streams, ribbed bedforms frequently form at localized areas of ice flow slowdown (i.e. sticky spots, Stokes et al., 2007, 2016) due to transitions and variations in the basal thermal regime and/or in the basal shear stress. Four major formation processes are proposed for ribbed bedformsTheoretical models of formation invoke either basal thermal conditions, meltwater flows or initial bed topography, to explain the shape and periodicity of ribbed bedforms. : Four categories of formation processes have been proposed: (i) overriding, deformation or reshaping of pre-existing sedimentary mounds , such as former streamlined or marginal landforms, by overriding ice (Boulton, 1987; Lundqvist, 1989; Möller, 2006); (ii) ice-fracturing and extension of frozen-bed along transitions fromof warm-to-cold ice bases, where tensional stresses increase (Hättestrand and Kleman, 1999; Sarala, 2006); (iii) subglacial meltwater floods responsible for the formation of inverted erosional marks at the ice base, infilled by sediments (Shaw, 2002); (iviii) till deformationfolding and thrusting of subglacial sediment , or sediment-rich basal ice in response to the flow of ice over bed heterogeneities resulting from variations in pore water pressure, basal thermal regime and bed strengthdewatered till and topographic relief (Terzaghi, 1931; Shaw, 1979; Bouchard, 1989; Kamb, 1991; Tulaczyk et al., 2000b; Lindén et al., 2008; Stokes et al., 2008), and (iv) in-situ generation of .The last process is consistent with physically-based mathematical models demonstrating that ribbed bedforms naturally ariseing from wavy-instabilities in the coupled combined flow of ice, basal meltwater and sediment viscoplastic till 
[revised manuscript text omitted]

[Figure]

**Commenté [J2]:** Figure 9 has been modified as mentioned in Responses to reviewers

[revised manuscript text omitted]